# Adaptive Data-Borrowing for Improving Treatment Effect Estimation using External Controls

**Qinwei Yang[1], Jingyi Li[2], and Peng Wu[1]***

Beijing Technology and Business University[1], National University of Singapore[2]

## Abstract

Randomized controlled trials (RCTs) often exhibit limited inferential efficiency in estimating treatment effects due to small sample sizes. In recent years, the combination of external controls has gained increasing attention as a means of improving the efficiency of RCTs. However, external controls are not always comparable to RCTs, and direct borrowing without careful evaluation can introduce substantial bias and reduce the efficiency of treatment effect estimation. In this paper, we propose a novel influence-based adaptive sample borrowing approach that effectively quantifies the "comparability" of each sample in the external controls using influence function theory. Given a selected set of borrowed external controls, we further derive a semiparametric efficient estimator under an exchangeability assumption. Recognizing that the exchangeability assumption may not hold for all possible borrowing sets, we conduct a detailed analysis of the asymptotic bias and variance of the proposed estimator under violations of exchangeability. Building on this bias-variance trade-off, we further develop a data-driven approach to select the optimal subset of external controls for borrowing. Extensive simulations and real-world applications demonstrate that the proposed approach significantly enhances treatment effect estimation efficiency in RCTs, outperforming existing approaches.

## 1 Introduction

Randomized controlled trials (RCTs) are regarded as the gold standard for estimating treatment effects, as randomization effectively eliminates confounding bias [1, 2, 3, 4]. However, their inferential efficiency is often limited by small sample sizes, due to high costs and lengthy recruitment periods [5, 6, 7, 8]. To address this challenge, there has been growing interest in novel clinical trial designs that leverage external real-world datasets containing only control arms, referred to as external controls, to improve treatment effect estimation in RCTs. For example, in digital marketing, platforms may combine small-scale A/B test data with historical single-arm user behavior logs to enhance inference [9]. In medical research, oncology trials often incorporate historical control data to strengthen causal conclusions [10]. Similarly, rare disease studies frequently augment small randomized trials with matched external controls from disease registries to improve statistical power [11].

To fully exploit external controls, most studies rely on the exchangeability assumption [12], which posits that the distribution of potential outcomes under control remains invariant between RCTs and external controls when conditioned on baseline covariates [13, 14, 15, 12, 16]. However, the exchangeability assumption requires that individuals in both RCTs and external controls follow the same pattern between the covariates and the potential outcome under control, which may be less plausible in real-world applications due to individual heterogeneity [17, 18, 19]. In practice, external controls usually have significantly larger sample sizes than RCTs and often exhibit greater individual heterogeneity [15, 12, 20]. When some individuals in the external controls display patterns that differ from those in the RCTs, the exchangeability assumption is violated and leads to biased conclusions.

---

*Corresponding author: pengwu@btbu.eud.cn

39th Conference on Neural Information Processing Systems (NeurIPS 2025).

*In this paper, we aim to improve treatment effect estimation in RCTs by adaptively borrowing external controls without relying on the exchangeability assumption. The key lies in (a) measuring the "comparability" of each sample in the external controls for treatment effect estimation, thereby distinguishing between comparable and non-comparable samples, and (b) defining and identifying the optimal set of comparable samples.*

*For the first goal (a)*, we propose an influence-based adaptive sample borrowing approach. Specifically, We employ the influence function [21, 22] to quantify how each external control sample perturbs the outcome model fitted on the RCT's control group, yielding influence scores that reflect the comparability of each sample in external controls. Intuitively, a sample with a smaller influence score has less impact on the outcome model in the RCT and is therefore considered more comparable. Including such samples in the RCT can effectively increase the control group's sample size without introducing bias. In contrast, a higher influence score suggests that including the sample would substantially affect the outcome model, thereby introducing bias into the treatment effect estimation. Using ranked influence scores, we construct nested candidate subsets of external control samples.

*For the second goal (b)*, we begin by deriving the semiparametric efficient estimator [23] that combines RCT data with an arbitrary candidate set of external control samples under the exchangeability assumption. Such an estimator minimizes the asymptotic variance among all regular estimators and is often considered optimal [24, 25] under the given assumptions. We establish the consistency and asymptotic normality of the proposed estimator. Then, recognizing that the exchangeability assumption may not hold for all candidate external control samples, we analyze the asymptotic bias and variance of the proposed estimator under violations of exchangeability. Finally, to determine the optimal external control samples for treatment effect estimation, we propose a data-driven approach that minimizes the estimator's mean squared error. The main contributions are summarized as follows.

- We reveal the limitations of existing approaches for estimating treatment effects by combining RCT data with external controls.

- We propose an influence-based sample borrowing approach, which can effectively quantify the comparability of each sample in the external controls.

- We develop a data-driven approach to select the optimal subset of external control samples based on the MSE of the proposed estimator.

- We conduct extensive experiments on both simulated and real-world datasets, demonstrating that the proposed approach outperforms the existing baseline approaches.

## 2   Related Work

There has been an increasing amount of research in settings where RCT data are augmented with external controls to improve efficiency in estimating treatment effects since [26], termed *external control*, *historical control*, or *history borrowing* in related literature. Under the exchangeability assumption (or its analog) for the potential outcome under control, several approaches [13, 15, 27] have been proposed to estimate treatment effects in RCTs using external controls. However, the exchangeability assumption may be less plausible in practice due to individual heterogeneity. As a result, combining the RCT data and external controls directly will lead to bias. To address the issue, various methods have been developed, including matching and bias adjustment [28], power priors [29], meta-analytic predictive priors [30], and conformal prediction [31]. More recently, a state-of-the-art approach proposed by [17] revealed the limitations of previous approaches and developed an adaptive lasso-based borrowing approach, which borrows a comparable subset of external controls through bias penalization. Unlike these studies, we propose a novel influence-based sample borrowing approach that utilizes influence functions to quantify the perturbation effect of each external control sample on the outcome model in the RCT controls. This enables the derivation of individual-level influence scores that capture the compatibility of external controls. Also, we develop approaches for borrowing the optimal subset of external controls. In this sense, our approach complements existing methodologies.

Beyond combining RCT data with external control data, there are many other settings for data combination [12, 16, 32, 33, 34]. For example, one can combine RCT (or experimental) data with external data that contain only covariates [35, 36]. In such settings, the goal is typically to generalize the causal effect from the RCT to the external population, rather than to improve causal effect estimation within the RCT itself. Another common setting involves combining RCT data with

confounded observational data that suffer from unobserved confounders [37, 38]. In addition, several studies have combined multiple datasets for causal inference [16, 39, 40]. When combining multiple datasets, it is important not only to consider which identifiability assumptions to adopt in order to improve causal effect estimation, but also to address how to preserve privacy. Unlike these studies that either use the entire external dataset or do not use external data at all, our method performs individual-level selection of external data.

## 3 Preliminaries

Let $X \in \mathcal{X} \subset \mathbb{R}^p$ denote the observed pre-treatment covariates and $Y \in \mathcal{Y} \subset \mathbb{R}$ the outcome of interest. The binary treatment indicator is $A \in \{0, 1\}$, where $A = 1$ indicates treatment and $A = 0$ indicates control. Under the potential outcomes framework [41, 42], each individual has two potential outcomes $Y(0)$ and $Y(1)$, corresponding to control and treatment, respectively. We maintain the stable unit treatment value assumption [43], the observed outcome is $Y = (1 - A)Y(0) + AY(1)$.

Suppose we have access to RCT data and external control data (external controls) containing only control samples. The RCT data and external controls are denoted by

$$\{X_i, A_i, Y_i, R_i = 1, i \in \mathcal{E}\} \text{ and } \{X_j, A_j = 0, Y_j, R_j = 0, j \in \mathcal{O}\},$$

where $R$ is a data source indicator: $R = 1$ corresponds to the RCT data $\mathcal{E}$ and $R = 0$ to the external controls $\mathcal{O}$. Let $N_{\mathcal{E}}$ and $N_{\mathcal{O}}$ denote the sample sizes of the RCT data and external controls, respectively. Let $\mathbb{P}(\cdot \mid R = 1)$ and $\mathbb{P}(\cdot \mid R = 0)$ represent the population distributions of the RCT data and external controls, respectively, and let $\mathbb{E}$ denote the expectation under $\mathbb{P}$.

The causal estimand of interest is the average treatment effect (ATE) in the RCT population, which is defined as $\tau = \mathbb{E}[Y(1) - Y(0) \mid R = 1]$. For identification, we invoke the common assumption in the causal inference [44, 45, 46].

**Assumption 1 (Strong Ignorability for RCT Data)** *(a)* $A \perp\!\!\!\perp \{Y(0), Y(1)\} \mid (X, R = 1)$; *(b)* $0 < e_1(x) \triangleq \mathbb{P}(A = 1 \mid X = x, R = 1) < 1$ *for all $x$, where $e_1(x)$ is the propensity score.*

Assumption 1(a) suggests that given the covariates $X$, treatment $A$ is independent of the potential outcome $Y(a)$. This implies that confounding between $A$ and $Y$ can be eliminated by conditioning on $X$. Assumption 1(b) ensures that individuals with $X = x$ have a positive probability of receiving treatment. Assumption 1 is inherently satisfied in RCTs due to the randomization mechanism. Under Assumption 1, $\tau$ can be identified based only on the RCT data,

$$\tau = \mathbb{E}[\mu_1(X) - \mu_0(X) \mid R = 1],$$

where $\mu_a(X) = \mathbb{E}[Y \mid X, A = a, R = 1]$ for $a = 0, 1$, are the outcome models in the RCT data.

When only RCT data are available, the semiparametric efficient estimator of $\tau$ under Assumption 1 is

$$\hat{\tau}_{\text{aipw}} = \frac{1}{N_{\mathcal{E}}} \sum_{i \in \mathcal{E}} \frac{A_i\{Y_i - \hat{\mu}_1(X_i)\}}{\hat{e}_1(X_i)} - \frac{(1 - A_i)\{Y_i - \hat{\mu}_0(X_i)\}}{1 - \hat{e}_1(X_i)} + \{\hat{\mu}_1(X_i) - \hat{\mu}_0(X_i)\},$$

where $\hat{\mu}_0(x)$, $\hat{\mu}_1(x)$, and $\hat{e}_1(x)$ are estimates of $\mu_0(x)$, $\mu_1(x)$, and $e_1(x)$, respectively. This is the classical augmented inverse probability weighting (AIPW) estimator, widely studied in prior work [47, 48, 49, 50, 51, 52, 53, 54]. The $\hat{\tau}_{\text{aipw}}$ serves as an asymptotically unbiased benchmark for $\tau$. Nevertheless, estimating $\tau$ using only the RCT data often lacks efficiency due to small sample sizes, resulting from high costs, long recruitment periods, and ethical or feasibility constraints. This paper aims to improve the estimation efficiency of $\tau$ by fully leveraging external controls.

## 4 Motivation

To leverage external controls, most studies [13, 14, 12] rely on the exchangeability assumption [55].

**Assumption 2 (Exchangeability)** $R \perp\!\!\!\perp Y(0) \mid X$ *in* $\mathbb{P}$.

Assumption 2 states that the conditional mean of $Y(0)$ is identical between the RCT data and external controls, i.e., $\mathbb{E}[Y(0) \mid X = x, R = 0] = \mathbb{E}[Y(0) \mid X = x, R = 1]$. This establishes a connection

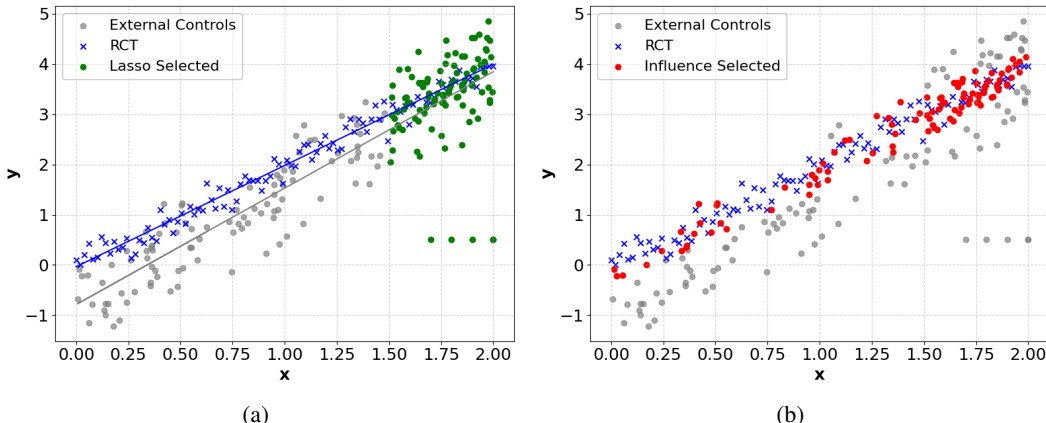

(a)                                                        (b)

Figure 1: Borrowing behavior comparison of different approaches in a synthetic example. (a) The borrowed samples of the adaptive lasso-based borrowing approach [17]. (b) The borrowed samples of the proposed influence-based borrowing approach. The influence-based approach identifies a set of samples with highly comparable. Details of the synthetic example can be found in Section 7.

between RCT data and external controls. Under this assumption, we can use the *entire* external controls to improve the treatment effect estimation in RCTs [15, 56].

However, the exchangeability assumption is usually implausible in practice due to individual heterogeneity. Thus, directly integrating external controls without proper scrutiny can lead to biased estimates. To mitigate this issue, a state-of-the-art approach has been proposed by Gao et al. (2025) [17] to find the optimal subset of external controls for integration to improve treatment effect estimation in RCTs. In this section, we provide a brief overview of this approach and analyze its limitations, thereby motivating this work.

Specifically, Gao et al. (2025) [17] introduced a vector of bias parameter $\boldsymbol{b}_0 = (b_{1,0}, \ldots, b_{N_{\mathcal{O}},0})$ for all $j \in \mathcal{O}$, where $b_{j,0} = \mathbb{E}(Y_j \mid X_j, R = 0) - \mathbb{E}(Y_j \mid X_j, A_j = 0, R = 1) := \mu_{0,\mathcal{O}}(X_j) - \mu_0(X_j)$. The bias parameter $\boldsymbol{b}_0$ quantifies the difference in conditional mean outcomes between external controls and RCT's controls. The authors treated external control samples with zero bias as "comparable" and aimed to identify such unbiased samples from the pool of external controls. Specifically, let $\hat{b}_{j,0} = \hat{\mu}_{0,\mathcal{O}}(X_j) - \hat{\mu}_0(X_j)$ be a consistent estimator for $b_{j,0}$ and let $\hat{\boldsymbol{b}} = (\hat{b}_{1,0}, \ldots, \hat{b}_{N_{\mathcal{O}},0})$ be an initial estimator for $\boldsymbol{b}_0$. Then, they obtained an sparse estimator of $\boldsymbol{b}_0$ via optimizing the adaptive lasso loss:

$$\tilde{\boldsymbol{b}} = \arg\min_{\boldsymbol{b}} \left\{ (\widehat{\boldsymbol{b}} - \boldsymbol{b})^{\mathrm{T}} \widehat{\Sigma}_{\boldsymbol{b}}^{-1} (\widehat{\boldsymbol{b}} - \boldsymbol{b}) + \lambda \sum_{j \in \mathcal{O}} \frac{|b_{j,0}|}{|\hat{b}_{j,0}|^{\nu}} \right\}, \tag{1}$$

where $\widehat{\Sigma}_{\boldsymbol{b}}$ is the variance estimate of $\hat{\boldsymbol{b}}$, $(\lambda, \nu)$ are two tuning parameters. Finally, they borrowed external control samples with estimated zero bias.

To better understand the borrowing behavior of the adaptive lasso-based approach, we conduct a simulation using a simple synthetic dataset (see Section 7 for details). As shown in Figure 1(a), where green points denote the external control samples borrowed by the adaptive lasso-based approach, we observe that this approach exhibits several limitations:

- Suboptimal comparability: As shown in Figure 1(a), the samples (marked in green) borrowed by the adaptive lasso-based approach may exhibit patterns that differ substantially from those in the RCT's controls, resulting in suboptimal comparability. Intuitively, the adaptive lasso-based approach borrows samples with small values of $b(x) := \mu_{0,\mathcal{O}}(x) - \mu_0(x)$, where $b(x)$ represents an average discrepancy that overlooks individual-level heterogeneity in the neighborhood of $x$. Consequently, the approach may include samples with high variability (even picking out several outliers), simply because their mean outcomes are close to those in RCT's controls.

- Sensitivity to outliers: The final estimator $\tilde{\boldsymbol{b}}$ heavily depends on the estimate of $\hat{\mu}_{0,\mathcal{O}}(x)$, which is learned using all samples from the external controls. When outliers are present in the external

controls, the estimate of $\hat{\mu}_{0,\mathcal{O}}(x)$ may become biased, introducing estimation errors for individual samples and exacerbating non-comparability issues.

These limitations greatly hinder the effective use of external controls, limiting their potential to improve the efficiency of treatment effect estimation in RCTs. Thus, developing an improved data borrowing methodology is essential.

## 5    Adaptive Sample Borrowing via Influence Function

In this section, we propose a novel influence-based borrowing approach designed to more effectively incorporate external controls, thereby enhancing the efficiency of treatment effect estimation.

The key lies in how to measure the comparability of each external control sample to the RCT's controls. We formalize this problem by asking the counterfactual: how do the parameters of the outcome model of the RCT's controls change if we add an external control sample? If the model parameters show minimal change, the sample can be considered comparable and suitable for borrowing.

Denote $\mathcal{C} = \{X_i, A_i = 0, Y_i, R_i = 1, i \in \mathcal{E}\}$ as the RCT's controls, and $N_\mathcal{C}$ is the sample size. Given RCT's controls $\{Z_i = (X_i, Y_i), Z_i \in \mathcal{C}\}$. To train the model $\hat{\mu}_0$ (parameterized by $\theta$), we learn the model parameters by the empirical risk minimizer, $\hat{\theta} \stackrel{\text{def}}{=} \arg\min_{\theta \in \Theta} \sum_{Z_i \in \mathcal{C}} L(Z_i; \theta)$ where $L(Z_i; \theta) = (Y_i - \hat{\mu}_0(X_i; \theta))^2$ that is twice-differentiable and convex in $\theta$, and $\lambda > 0$ controls regularization strength. For an added external control sample $z = (x, y)$, we denote the modified parameters as $\hat{\theta}_{+z}$, which is obtained by retraining the model after adding $z$ as $\hat{\theta}_{+z} \stackrel{\text{def}}{=} \arg\min_{\theta \in \Theta} \sum_{Z_i \in \mathcal{C} \cup z} L(Z_i; \theta)$. Then, we can obtain the influence score of the external control sample $z$ on loss over RCT's controls,

$$\mathcal{IF}_{\text{loss}}(z) \stackrel{\text{def}}{=} \sum_{Z_i \in \mathcal{C}} |L(Z_i, \hat{\theta}_{+z}) - L(Z_i, \hat{\theta})|, \tag{2}$$

The influence score $\mathcal{IF}_{\text{loss}}(z)$ measures the actual influence of $z$. A large value of $\mathcal{IF}_{\text{loss}}$ indicates that $z$ has a significant impact on the model $\hat{\mu}_0$.

However, retraining the model for each added $z$ is prohibitively slow and expensive. Fortunately, instead of retraining the model, the *influence function* gives an approximation of how much the model changes when external control samples are added to the RCT data [57]. The main idea behind the influence function is to weight the external control sample $z$ by infinitesimal steps $\epsilon$ to produce new model parameters $\hat{\theta}_{\epsilon,z} \stackrel{\text{def}}{=} \arg\min_{\theta \in \Theta} N_\mathcal{C}^{-1} \sum_{Z_i \in \mathcal{C}} L(Z_i, \theta) + \epsilon L(z, \theta)$. A classic result [22, 57] tells us that the influence of upweighting $z$ on the parameters $\hat{\theta}$ is given by

$$\mathcal{I}_{\text{params}}(z) \stackrel{\text{def}}{=} \left. \frac{d\hat{\theta}_{\epsilon,z}}{d\epsilon} \right|_{\epsilon=0} = -H_{\hat{\theta}}^{-1} \nabla_\theta L(z, \hat{\theta}), \tag{3}$$

where $H_{\hat{\theta}} \stackrel{\text{def}}{=} N_\mathcal{C}^{-1} \sum_{Z_i \in \mathcal{C}} \nabla_{\hat{\theta}}^2 L(Z_i, \hat{\theta})$ is the Hessian matrix. Then, the influence of adding an external control sample $z$ on the prediction loss of a single sample $Z_i \in \mathcal{C}$ can be expressed in a closed-form expression by applying the chain rule, as follows,

$$\mathcal{I}_{\text{loss}}(z, Z_i) \stackrel{\text{def}}{=} \left. \frac{dL(Z_i, \hat{\theta}_{\epsilon,z})}{d\epsilon} \right|_{\epsilon=0} = \nabla_\theta L(Z_i, \hat{\theta})^\top \left. \frac{d\hat{\theta}_{\epsilon,z}}{d\epsilon} \right|_{\epsilon=0} = -\nabla_\theta L(Z_i, \hat{\theta})^\top H_{\hat{\theta}}^{-1} \nabla_\theta L(z, \hat{\theta}). \tag{4}$$

According to Eq. (4), $\mathcal{I}_{\text{loss}}(z, Z_i)$ is equivalent to "the first-order derivatives of the model parameters $\hat{\theta}$ at $Z_i$" multiplied by "the parameter change when we add the external control sample $z$". Furthermore, we can quantify the total influence of an external control sample $z$ on the entire RCT's controls as the influence score, which is given as,

$$\mathcal{IF}_{\text{loss}}(z) \approx \sum_{Z_i \in \mathcal{C}} \left| \nabla_\theta L(Z_i, \hat{\theta})^\top H_{\hat{\theta}}^{-1} \nabla_\theta L(z, \hat{\theta}) \right| \tag{5}$$

For all external control samples $\{Z_j = (X_j, Y_j), j \in \mathcal{O}\}$, we can calculate the influence score set $\{\mathcal{IF}_{\text{loss}}(Z_j), j \in \mathcal{O}\}$ according to Eq. (5). The influence scores measure the comparability of each

external control sample. **The proposed adaptive data-borrowing method is based on the influence scores and consists of two main steps:**

**Step 1.** Based on the ranking of influence scores, we construct a series of nested subsets of external controls, where each subset comprises the top-$K$ most comparable samples (i.e., the top-$K$ samples with the smallest influence scores).

**Step 2.** Find the optimal $K$ that minimizes the mean square error (MSE) of the estimator based on the RCT data and the selected top-$K$ external controls. This method will be detailed in Section 6.

*Unlike the adaptive lasso-based approach, the influence-based approach does not rely on modeling $\mu_{0,\mathcal{O}}(x)$. In addition, it is noteworthy that the influence score is defined at the individual level, and the influence score for each point is unaffected by other points in the external controls.* As a result, it is robust to outliers in external controls. As shown in Figure 1(b), the proposed method effectively selects the top-$K$ points (in red) that are close to the RCT controls, while remaining robust to outliers in the external controls.

## 6 Improved Estimation by Leveraging of Borrowed External Controls

In this section, based on the ranking of influence scores, our goal is to select the optimal subset of external controls that minimizes the mean squared error (MSE) of the proposed estimator.

Let $\mathcal{S} \subseteq \mathcal{O}$ be a subset of $\mathcal{O}$ borrowed by the influence function, and $N_{\mathcal{S}}$ is the sample size. For ease of presentation, we denote $\mathbb{P}_{\mathcal{S}}$ as the combined population that $\mathbb{P}_{\mathcal{S}}(\cdot \mid R = 1)$ and $\mathbb{P}_{\mathcal{S}}(\cdot \mid R = 0)$ denote the distributions of RCT data and borrowed external controls. The expectation operator of $\mathbb{P}_{\mathcal{S}}$ is denoted by $\mathbb{E}_{\mathcal{S}}$. For clarity, we summarize the nuisance parameters in Table 1 that are utilized in the following theory analysis, and all of them can be identified from the observed data.

In subsection 6.1, we construct the estimator of $\tau$ by combining the RCT data $\mathcal{E}$ and the borrowed set $\mathcal{S}$, under the weaker version of Assumption 2 (Assumption 3 below). In subsection 6.2, we first analyze the bias and variance of the estimator under the violation of Assumption 3, and then select the optimal $K$ that minimizes the corresponding MSE.

### 6.1 Fused Estimator Based on Exchangeability for Borrowed External Controls

To fully utilize the borrowed data, we aim to derive the semiparametric efficient estimator of $\tau$, which is regarded as optimal since it achieves the semiparametric efficiency bound—yielding the smallest asymptotic variance under standard regularity conditions [23, 24]. To construct the semiparametric efficient estimator, we typically first derive the efficient influence function and the semiparametric efficiency bound based on the available data and assumptions.

**Assumption 3 (Exchangeability for Borrowed External Controls)** $R \perp\!\!\!\perp Y(0) \mid X$ *in* $\mathbb{P}_{\mathcal{S}}$.

Assumption 3 is analogous to, but weaker than, Assumption 2, differing only by replacing $\mathbb{P}$ with $\mathbb{P}_{\mathcal{S}}$. This implies that the outcome regression functions for $Y(0)$ given covariates are the same in $\mathcal{E}$ and $\mathcal{S}$, that is, $\mathbb{E}_{\mathcal{S}}[Y(0) \mid X, R = 1] = \mathbb{E}_{\mathcal{S}}[Y(0) \mid X, R = 0]$. Assumption 3 is plausible when $\mathcal{S}$ is appropriately selected by the influence function. As shown in Figure 1(b), the RCT controls (blue points) and the selected external controls (red points) are close, and follow a similar distribution, supporting the plausibility of Assumption 3.

**Lemma 1** *Under Assumptions 1 and 3, the efficient influence function of $\tau$ is*

$$\phi = \frac{\pi(X)}{q}\left\{ \frac{RA(Y - m_1(X))}{e_{\mathcal{S}}(X)} - \frac{(1-A)(Y - m_0(X))}{1 - e_{\mathcal{S}}(X)} \right\} + \frac{R}{q}\{m_1(X) - m_0(X) - \tau\},$$

*where $e_{\mathcal{S}}(X)$, $\pi(X)$, $m_1(X)$, and $m_0(X)$ are defined in Table 1 and $q = \mathbb{P}_{\mathcal{S}}(R = 1)$. The associated semiparametric efficiency bound for $\tau$ is* $\mathrm{Var}(\phi)$.

Lemma 1 presents the efficient influence function of $\tau$ under Assumptions 1 and 3. Based on it, we can construct the estimator of $\tau$ as follows

$$\hat{\tau}_{\mathcal{S}} = \frac{1}{N_{\mathcal{E}} + N_{\mathcal{S}}} \sum_{i \in \mathcal{E} \cup \mathcal{S}} \frac{\hat{\pi}(X_i)}{q}\left\{ \frac{R_i A_i(Y_i - \hat{m}_1(X_i))}{\hat{e}_{\mathcal{S}}(X_i)} - \frac{(1 - A_i)(Y_i - \hat{m}_0(X_i))}{1 - \hat{e}_{\mathcal{S}}(X_i)} \right\} + \frac{R_i}{q}\{\hat{m}_1(X_i) - \hat{m}_0(X_i)\}.$$

Table 1: Nuisance parameters in $\hat{\tau}_{\text{aipw}}$ and $\hat{\tau}_{\mathcal{S}}$.

| Nuisance parameters | Description |
|---|---|
| $e_1(X) = \mathbb{P}(A = 1 \mid X, R = 1) = \mathbb{P}_{\mathcal{S}}(A = 1 \mid X, R = 1),$ | propensity score in $\mathcal{E}$ |
| $\mu_a(X) = \mathbb{E}(Y \mid X, A = a, R = 1) = \mathbb{E}_{\mathcal{S}}(Y \mid X, A = a, R = 1)$ | outcome regression function in $\mathcal{E}$ |
| $\pi(X) = \mathbb{P}_{\mathcal{S}}(R = 1 \mid X)$ | sampling score in $\mathcal{E} \cup \mathcal{S}$ |
| $e_{\mathcal{S}}(X) = \mathbb{P}_{\mathcal{S}}(A = 1 \mid X) = e_1(X)\pi(X),$ | propensity score in $\mathcal{E} \cup \mathcal{S}$ |
| $m_a(X) = \mathbb{E}_{\mathcal{S}}[Y \mid X, A = a],$ | outcome regression function in $\mathcal{E} \cup \mathcal{S}$ |

Note that by definition, $\mu_1(X) = m_1(X)$ but $\mu_0(X) \neq m_0(X)$.

where $\hat{\pi}(x), \hat{e}_1(x), \hat{m}_a(x)$ are estimates of $\pi(x), e_1(x), m_a(x)$ for $a = 0, 1$, and $\hat{e}_{\mathcal{S}}(x) = \hat{e}_1(x)\hat{\pi}(x)$.

**Lemma 2** *Under Assumptions 1 and 3, if* $||\hat{e}_1(x) - e_1(x)||_2 \cdot ||\hat{m}_a(x) - m_a(x)||_2 = o_{\mathbb{P}}(n^{-1/2})$ *and* $||\hat{\pi}(x) - \pi(x)||_2 \cdot ||\hat{m}_a(x) - m_a(x)||_2 = o_{\mathbb{P}}(n^{-1/2})$ *for all* $x \in \mathcal{X}$ *and* $a \in \{0, 1\}$, *then* $\hat{\tau}_{\mathcal{S}}$ *satisfies* $\sqrt{N_{\mathcal{E}} + N_{\mathcal{S}}}(\hat{\tau}_{\mathcal{S}} - \tau) \xrightarrow{d} \mathcal{N}(0, \sigma^2)$, *where* $\sigma^2$ *is the semiparametric efficiency bound of* $\tau$, *and* $\xrightarrow{d}$ *means convergence in distribution.*

Lemma 2 establishes the consistency and asymptotic normality of the estimator $\hat{\tau}_{\mathcal{S}}$. In addition, it shows that $\hat{\tau}_{\mathcal{S}}$ is semiparametric efficient, provided that the nuisance parameters are estimated at a convergence rate faster than $n^{-1/4}$. These conditions are common in causal inference and are easily satisfied using a variety of flexible machine learning methods [58].

## 6.2 Find the Optimal Borrowed Set

In Section 6.1, given a borrowed set $\mathcal{S}$, we construct the semiparametric efficient estimator $\hat{\tau}_{\mathcal{S}}$ by combining data from $\mathcal{E}$ and $\mathcal{S}$. However, the proposed estimator $\hat{\tau}_{\mathcal{S}}$ relies on the key Assumption 3, which may not hold for all choices of $\mathcal{S}$.

When Assumption 3 is violated, $\hat{\tau}_{\mathcal{S}}$ is no longer a consistent estimator of $\tau$ and yields a bias. Intuitively, selecting $\mathcal{S}$ involves a bias-variance trade-off: increasing the sample size in $\mathcal{S}$ tends to reduce variance but may also introduce "non-comparable" samples, thereby increasing bias. The proposed approach is based on the bias-variance analysis of $\hat{\tau}_{\mathcal{S}}$.

**Theorem 1 (Bias-Variance Analysis)** *Under Assumption 1 only, if* $||\hat{e}_1(x) - e_1(x)||_2 \cdot ||\hat{m}_a(x) - m_a(x)||_2 = o_{\mathbb{P}}(n^{-1/2})$ *and* $||\hat{\pi}(x) - \pi(x)||_2 \cdot ||\hat{m}_a(x) - m_a(x)||_2 = o_{\mathbb{P}}(n^{-1/2})$ *for all* $x \in \mathcal{X}$ *and* $a \in \{0, 1\}$, *then* $\hat{\tau}_{\mathcal{S}}$ *satisfies* $\sqrt{N_{\mathcal{E}} + N_{\mathcal{S}}}\{\hat{\tau}_{\mathcal{S}} - \tau - \text{bias}(\hat{\tau}_{\mathcal{S}})\} \xrightarrow{d} \mathcal{N}(0, \sigma^2)$, *where* $\sigma^2 = \text{Var}(\phi)$, $\phi$ *is defined in Proposition 1, and* $\text{bias}(\hat{\tau}_{\mathcal{S}}) = \mathbb{E}_{\mathcal{S}}\left[\frac{R}{q}\{\mu_0(X) - m_0(X)\}\right]$.

Theorem 1 extends Lemma 2 by allowing Assumption 3 to be violated. When Assumption 3 holds, the bias term vanishes, and Theorem 1 reduces to Lemma 2, with the asymptotic variance $\sigma^2$ achieving the semiparametric efficiency bound for $\tau$ under Assumptions 1–3. In contrast, when Assumption 3 is violated, bias arises and is proportional to the expectation of $\mu_0(X) - m_0(X)$, and the asymptotic variance $\sigma^2$ retains the same form but is no longer the semiparametric efficiency bound.

Based on Theorem 1, we propose to select the optimal $\mathcal{S}$ that minimizes the MSE of $\hat{\tau}_{\mathcal{S}}$ from the candidate sets when Assumption 3 is violated. Specifically, the MSE of the estimator $\hat{\tau}_{\mathcal{S}}$ is $\text{MSE}(\hat{\tau}_{\mathcal{S}}) = \text{bias}^2(\hat{\tau}_{\mathcal{S}}) + \text{var}(\hat{\tau}_{\mathcal{S}})$. We estimate the bias using $\hat{\tau}_{\mathcal{S}} - \hat{\tau}_{\text{aipw}}$ and estimate the variance using the sample variance of $\hat{\phi}$, where $\hat{\phi}$ is the estimate of $\phi$ obtained by substituting the nuisance parameters with their estimated values. Our goal is to find the optimal subset $\mathcal{S}$ defined by

$$\mathcal{S}^* = \arg\min_{\mathcal{S}_k \in \mathbf{S}} \text{MSE}(\hat{\tau}_{\mathcal{S}_k}),$$

where $\mathbf{S} = \{\mathcal{S}_k : k = 1, ..., N_{\mathcal{O}}\}$, $\mathcal{S}_k$ denotes a subset of external controls corresponding to the top-$k$ smallest influence scores, $N_{\mathcal{O}}$ is the sample size of external controls.

**Importantly**, the proposed selection strategy for $\mathcal{S}^*$ does not rely on Assumption 2 or 3, nor does it impose any restrictions on the distribution of the external controls. Therefore, our method exhibits high applicability across a variety of settings. RCT data typically have relatively small sample sizes, as their collection is often time-consuming and costly, and further limited by the inclusion

criteria set by the experimenter [59]. In comparison, external controls are observational, more readily available, and often drawn from larger and more diverse sources, exhibiting greater individual heterogeneity [15, 20, 60]. As a result, it is inevitable that some individuals (or outliers) in the external controls will exhibit patterns that differ significantly from those of the RCT controls. In this case, the adaptive lasso-based approach can only achieve sub-optimal performance (as discussed in Section 4), whereas our proposed method can effectively accommodate such scenarios.

# 7  A Synthetic Example

To better analyze the borrowing behavior of the proposed influence-based borrowing approach, we first compare it with the adaptive lasso-based borrowing approach [17] on a simple synthetic dataset.

$$Y_{\text{rt}} = 2X_{\text{rt}} + \epsilon_{\text{rt}}, \quad Y_{\text{ec}} = -1 + 2.5X_{\text{ec}} + \epsilon_{\text{ec}},$$

where $X_{\text{rt}}, X_{\text{ec}} \sim U(0, 2)$, and $\epsilon_{\text{rt}} \sim N(0, 0.2^2)$, $\epsilon_{\text{ec}} \sim N(0, 0.5^2)$. The $X_{\text{rt}}, X_{\text{ec}}$ is the 1-dimensional covariates. And the external controls contain five outlier samples.

As shown in Figure 1, the adaptive lasso-based approach proposed by [17] achieves only suboptimal comparability and is sensitive to outliers. This suggests that relying solely on the conditional mean difference may not be an optimal strategy for identifying comparable samples. In contrast, the proposed influence-based borrowing approach achieves better performance by employing the influence function theory to quantify the comparability of each external control.

# 8  Experiments

To demonstrate the effectiveness of the proposed approaches, we conduct experiments on three datasets, including two synthetic datasets and a real-world dataset. Experimental details (e.g., parameter settings), are provided in the Appendix C.

## 8.1  Experimental Setup

**Simulation Study.** We generate synthetic datasets to mimic both RCTs and external controls, with each observation characterized by $d = 8$ dimensional covariates $X$. The synthetic datasets consist of $N_{\mathcal{E}} = 400$ samples with $N_t = 300$ in the treated group and $N_c = 100$ in the control group, while the external control dataset includes $N_{\mathcal{O}} = 800$ samples. The treatment assignment $A$ for the RCTs is completely at random. Following [17], we designed two data-generating mechanisms: a linear outcome model (denoted as "Linear") and a nonlinear outcome model (denoted as "Nonlinear").

As outlined in Table 2, for the "Linear" mechanism, we simulate the RCTs baseline covariates $X_{\text{rt}} \sim N(\mu_1, \sigma_1^2)$ and the external controls baseline covariates $X_{\text{ec}} \sim N(\mu_2, \sigma_2^2)$. Here, $\mu_1 \neq \mu_2$, with $|\mu_2 - \mu_1|$ quantifying the magnitude of covariate shift between the two groups. The standard deviation $\sigma_2 > \sigma_1$ reflects greater heterogeneity in the external controls compared to the RCTs. We initially set $\mu_1 = 0, \sigma_1 = 1$ and $\mu_2 = 0.1, \sigma_2 = 2$. The coefficient $\Delta\beta_1 \sim U([0.8, 1.2]^d)$ characterizes the structural differences in outcome models between the external controls and the RCTs. $\delta$ and $T$ constitute the effect of concurrency bias, in which $T_i, i \in \mathcal{O}$, is simulated by taking values of $(0, 1, 2)$ with probability $1/3$ and $\delta$ represents the level of inconcurrency, where we set $\delta = 0.1$. Similarly, for the "Nonlinear" mechanism, $\tilde{X}_{\text{rt}} \sim N_{[-2,2]}(\mu_1, \sigma_1^2)$ and $\tilde{X}_{\text{ec}} \sim N_{[-4,4]}(\mu_2, \sigma_2^2)$, both follow a truncated normal distribution. The $\Delta\beta_2$ characterizes the structural differences and $\tilde{\delta}$ represents the level of inconcurrency. We set $\Delta\beta_2 \sim U([0.8, 1.2]^d)$ and $\tilde{\delta} = 1.0$.

Table 2: Simulation settings: model choices (linear and nonlinear).

| | generate mechanism | parameters |
|---|---|---|
| Linear | $Y_{rt} = \beta_1^T X_{\text{rt}} + A\alpha_1^T(1, X_{\text{rt}}) + \epsilon_{rt},$ 
 $Y_{ec} = (\beta_1 \cdot \Delta\beta_1)^T X_{\text{ec}} + \delta \cdot T + \epsilon_{ec},$ | $\beta_1 \sim U([-1,1]^d), \epsilon_{rt} \sim N(0, 1.0^2)$ 
 $\epsilon_{ec} \sim N(0, 1.5^2)$ |
| Nonlinear | $Y_{rt} = a \cdot \exp\{\beta_2^T \tilde{X}_{\text{rt}} + A\alpha_2^T(1, \tilde{X}_{\text{rt}})\} + \tilde{\epsilon}_{rt},$ 
 $Y_{ec} = a \cdot \exp\{(\beta_2 \cdot \Delta\beta_2)^T \tilde{X}_{\text{ec}}\} + \tilde{\delta} \cdot T + \tilde{\epsilon}_{ec},$ | $\beta_2 \sim U([-1,1]^d), \tilde{\epsilon}_{rt} \sim N(0, 1.0^2)$ 
 $\tilde{\epsilon}_{ec} \sim N(0, 2.0^2)$ |

**Real-Data Application.** In addition to synthetic datasets, we also utilize two real-world datasets from the National Supported Work (NSW) program (RCTs) [61] and the Population Survey of Income Dynamics (PSID) (external controls) [62]². The NSW dataset consists of $N_{\mathcal{E}} = 345$ samples with $N_t = 185$ in the treated group and 260 in the control group, where we randomly selected $N_c = 80$ samples as the control group, while the PSID dataset includes $N_{\mathcal{O}} = 123$ samples. The NSW program investigates whether providing intensive job training and supported work experience could improve employment outcomes for economically disadvantaged populations. This dataset contains a treatment indicator (i.e., the training program, A: 1 for treated, 0 otherwise), demographic covariates ( age, education years, race indicators), Socioeconomic variables (marital status, education attainment), and earnings (1974, 1975, and 1978 outcomes). We take the earnings in 1978 as the interesting outcome $Y$. The external control dataset, PSID, is used for external comparison with the NSW dataset, which includes the same 10 columns as the NSW dataset, with all observations representing untreated individuals (A = 0 for all individuals).

**Baselines and Evaluation metrics.** We compare our proposed influence-based approach ($\hat{\tau}_{\mathrm{if}}$) with the following baselines in the above datasets. (a) the augmented inverse probability weighting (AIPW) estimator without borrowing ($\hat{\tau}_{\mathrm{aipw}}$) [63] (based only on the RCT data); (b) the AIPW estimator with full borrowing ($\hat{\tau}_{\mathrm{full}}$) [15]. (c) the AIPW estimator with adaptive lasso-based borrowing approach ($\hat{\tau}_{\mathrm{lasso}}$) [17]. We report three evaluation metrics, including standard deviation (std), bias, and mean squared errors (MSE), with detailed definitions provided in Section 6.2.

## 8.2 Results

The adaptive lasso-based approach assesses the comparability of external control samples based on the magnitude of bias **b**, where a smaller bias indicates stronger comparability, and vice versa. In contrast, the influence-based approach evaluates comparability using influence scores, with smaller scores reflecting stronger comparability. By ranking bias **b**, we can identify the Top-K most comparable external control samples borrowed by the adaptive lasso-based approach. Similarly, by ranking influence scores, we can identify the Top-K most comparable external control samples borrowed by the proposed influence-based approach.

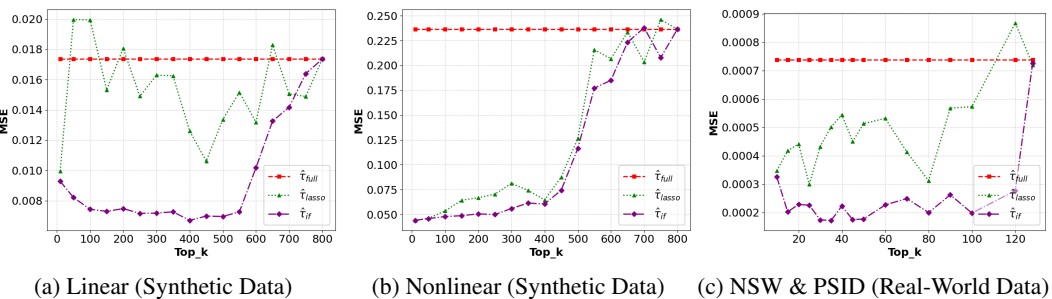

(a) Linear (Synthetic Data)  (b) Nonlinear (Synthetic Data)  (c) NSW & PSID (Real-World Data)

Figure 2: Comparison of performance of different approaches at different Top-K.

**Sample Borrowing Behavior Analysis**. Figure 2 presents the changes in MSE when borrowing different Top-K samples. First, we can find that the full borrowing approach $\hat{\tau}_{\mathrm{full}}$ is invariant across Top-K. It has a larger MSE than other approaches as it integrates the biased external controls for estimation. Second, the proposed $\hat{\tau}_{\mathrm{if}}$ consistently achieves a lower MSE than the adaptive lasso-based approach $\hat{\tau}_{\mathrm{lasso}}$. This is attributed to the better comparability of the external control samples borrowed by the proposed approach. Third, increasing Top-K leads to an initial decrease (or flattening) in MSE, followed by an upward trend, which indicates that our approach effectively prioritizes the more comparable external control samples, compared to the adaptive lasso-based approach.

**Performance Comparison.** To thoroughly evaluate the validity of the proposed approach, we present detailed experimental results in Table 3, including estimated standard deviations and biases across different Top-K values. We can find the following conclusion. First, due to the biases of the external controls, the full borrowing estimator $\hat{\tau}_{\mathrm{full}}$ is significantly different from $\hat{\tau}_{\mathrm{aipw}}$, leading to

---

²This data is available at `https://users.nber.org/~rdehejia/nswdata2.html`

Table 3: Comparison of our approach ($\hat{\tau}_{\text{if}}$), $\hat{\tau}_{\text{aipw}}$, $\hat{\tau}_{\text{full}}$ and $\hat{\tau}_{\text{lasso}}$ on three datasets, with standard deviation (std) and |bias| as evaluation metrics.

| Linear | Top-K=10 | | Top-K=50 | | Top-K=100 | | Top-K=150 | | Top-K=200 | | Top-K=250 | | Top-K=300 | |
|---|---|---|---|---|---|---|---|---|---|---|---|---|---|---|
| | std | \|bias\| | std | \|bias\| | std | \|bias\| | std | \|bias\| | std | \|bias\| | std. | \|bias\| | std | \|bias\| |
| $\hat{\tau}_{\text{aipw}}$ | 0.1075 | - | 0.1075 | - | 0.1075 | - | 0.1075 | - | 0.1075 | - | 0.1075 | - | 0.1075 | - |
| $\hat{\tau}_{\text{full}}$ | 0.0912 | 0.0951 | 0.0912 | 0.0951 | 0.0912 | 0.0951 | 0.0912 | 0.0951 | 0.0912 | 0.0951 | 0.0912 | 0.0951 | 0.0912 | 0.0951 |
| $\hat{\tau}_{\text{lasso}}$ | 0.0994 | 0.0082 | 0.1032 | 0.0964 | 0.1031 | 0.0965 | 0.1022 | 0.0698 | 0.1039 | 0.0851 | 0.1023 | 0.0665 | 0.1006 | 0.0784 |
| $\hat{\tau}_{\text{if}}$ | 0.0963 | **0.0005** | 0.0902 | **0.0075** | 0.0841 | **0.0185** | 0.0819 | **0.0240** | 0.0808 | **0.0307** | 0.0807 | **0.0252** | 0.0808 | **0.0251** |

| Nonlinear | Top-K=10 | | Top-K=50 | | Top-K=100 | | Top-K=150 | | Top-K=200 | | Top-K=250 | | Top-K=300 | |
|---|---|---|---|---|---|---|---|---|---|---|---|---|---|---|
| | std | \|bias\| | std | \|bias\| | std | \|bias\| | std | \|bias\| | std | \|bias\| | std | \|bias\| | std | \|bias\| |
| $\hat{\tau}_{\text{aipw}}$ | 0.2232 | - | 0.2232 | - | 0.2232 | - | 0.2232 | - | 0.2232 | - | 0.2232 | - | 0.2232 | - |
| $\hat{\tau}_{\text{full}}$ | 0.1862 | 0.4492 | 0.1862 | 0.4492 | 0.1862 | 0.4492 | 0.1862 | 0.4492 | 0.1862 | 0.4492 | 0.1862 | 0.4492 | 0.1862 | 0.4492 |
| $\hat{\tau}_{\text{lasso}}$ | 0.2093 | **0.0130** | 0.2124 | 0.0318 | 0.2218 | 0.0648 | 0.2236 | 0.1192 | 0.2202 | 0.1350 | 0.2270 | 0.1370 | 0.2280 | 0.1709 |
| $\hat{\tau}_{\text{if}}$ | 0.2090 | 0.0179 | 0.2117 | **0.0312** | 0.2136 | **0.0459** | 0.2153 | **0.0485** | 0.2165 | **0.0602** | 0.2152 | **0.0595** | 0.2128 | **0.1028** |

| NSW & PSID | Top-K=10 | | Top-K=15 | | Top-K=20 | | Top-K=25 | | Top-K=30 | | Top-K=35 | | Top-K=40 | |
|---|---|---|---|---|---|---|---|---|---|---|---|---|---|---|
| | std | \|bias\| | std | \|bias\| | std | \|bias\| | std | \|bias\| | std | \|bias\| | std | \|bias\| | std | \|bias\| |
| $\hat{\tau}_{\text{aipw}}$ | 0.0134 | - | 0.0134 | - | 0.0134 | - | 0.0134 | - | 0.0134 | - | 0.0134 | - | 0.0134 | - |
| $\hat{\tau}_{\text{full}}$ | 0.0149 | 0.0226 | 0.0149 | 0.0226 | 0.0149 | 0.0226 | 0.0149 | 0.0226 | 0.0149 | 0.0226 | 0.0149 | 0.0226 | 0.0149 | 0.0226 |
| $\hat{\tau}_{\text{lasso}}$ | 0.0142 | **0.0119** | 0.0144 | 0.0144 | 0.0144 | 0.0152 | 0.0136 | 0.0106 | 0.0140 | 0.0152 | 0.0140 | 0.0174 | 0.0145 | 0.0181 |
| $\hat{\tau}_{\text{if}}$ | 0.0131 | 0.0123 | 0.0127 | **0.0063** | 0.0128 | **0.0078** | 0.0130 | **0.0074** | 0.0129 | **0.0026** | 0.0130 | **0.0013** | 0.0132 | **0.0069** |

a significantly larger bias. Therefore, we exclude it from further comparisons below. Second, the proposed $\hat{\tau}_{\text{if}}$ is closer to the benchmark $\hat{\tau}_{\text{aipw}}$ with smaller bias compared with all baselines. Third, the proposed $\hat{\tau}_{\text{if}}$ has a smaller standard deviation than both $\hat{\tau}_{\text{lasso}}$ and $\hat{\tau}_{\text{aipw}}$. This is attributed to the high-quality external control samples borrowed by the proposed approach.

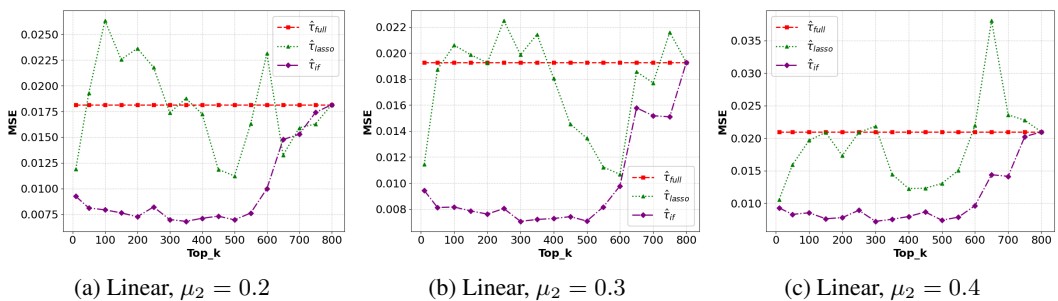

(a) Linear, $\mu_2 = 0.2$    (b) Linear, $\mu_2 = 0.3$    (c) Linear, $\mu_2 = 0.4$

Figure 3: Comparison of the performance of different approaches at different covariate shifts.

**Sensitivity Analysis.** We also conducted experiments to explore the robustness of the $\hat{\tau}_{\text{if}}$ to different $\mu_2$ (i.e., covariate shift), where we consider $\mu_2 = \{0.2, 0.3, 0.4\}$. As shown in Figure 3, we observe that: 1) the proposed $\hat{\tau}_{\text{if}}$ consistently outperforms the adaptive lasso-based approach; 2) as Top-K increases, MSE tends to decrease and then increase. In addition, we conducted experiments on different RCT control group sizes. We retain the size of the treatment group, but create different sub-samples by randomly selecting $N_c^s$ samples from its control group of RCT. In our simulation study, we consider control group sizes of $N_c^s \in \{70, 80, 90\}$, while for the real-data application, we examine a range of $N_c^s \in \{70, 75, 85\}$. The corresponding results are presented in Appendix B.1. These results further demonstrate the robustness of the proposed approach $\hat{\tau}_{\text{if}}$.

## 9   Conclusion

In this paper, we first reveal the limitations of existing approaches in borrowing external control samples for integration in estimating treatment effects. Then, we propose an influence-based borrowing approach that can overcome these limitations. The proposed approach consists of two key steps: measuring the "comparability" of each external control sample and identifying the optimal set of comparable samples. For the first step, we employ the influence function to obtain sample-specific influence scores. For the second step, we propose a data-driven approach to select the optimal set by minimizing the estimator's mean squared error (MSE). A limitation of this work is that computing the Hessian matrix for the influence function may have a high computational burden for large-scale models, particularly deep neural networks with millions or even billions of parameters.

## Acknowledgments and Disclosure of Funding

The authors thank Professor Shu Yang of North Carolina State University for her insightful comments and constructive discussions on this work. This research was supported the National Natural Science Foundation of China (No. 12301370), the BTBU Digital Business Platform Project by BMEC, the Beijing Key Laboratory of Applied Statistics and Digital Regulation, Academy for Interdisciplinary Studies at Beijing Technology and Business University, and the BTBU Research Foundation for Youth Scholars (No. BRFYS2025).

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

# A Technical Proofs

**Lemma 1.** Under Assumptions 1 and 3, the efficient influence function of $\tau$ is

$$\phi = \frac{\pi(X)}{q}\left\{\frac{RA(Y - m_1(X))}{e_\mathcal{S}(X)} - \frac{(1 - A)(Y - m_0(X))}{1 - e_\mathcal{S}(X)}\right\} + \frac{R}{q}\{m_1(X) - m_0(X) - \tau\},$$

where $e_\mathcal{S}(X)$, $\pi(X)$, $m_1(X)$, and $m_0(X)$ are defined in Table 1 of the main text. The associated semiparametric efficiency bound for $\tau$ is $\text{Var}(\phi)$.

**Proof of Lemma 1.** Recall that $e_\mathcal{S}(X) = e_1(X)\pi(X)$, $\pi(X) = \mathbb{P}_\mathcal{S}(R = 1 \mid X)$, $m_1(X) = \mathbb{E}_\mathcal{S}[Y \mid X, A = 1]$, $m_0(X) = \mathbb{E}_\mathcal{S}[Y \mid X, A = 0]$, and $e_1(X) = \mathbb{P}(A = 1 \mid X, R = 1) = \mathbb{P}_\mathcal{S}(A = 1 \mid X, R = 1)$ is the propensity score in the RCT data.

Let $f(x)$, $f(x|r = 1)$, and $f(x|r = 0)$ represent the density functions of $X$ in the combined data $\mathbb{P}_\mathcal{S}$, RCT data $\mathbb{P}_\mathcal{S}(\cdot|R = 1)$ and selected external data $\mathbb{P}_\mathcal{S}(\cdot|R = 0)$, respectively. Denote $f(y_0, y_1|x)$ as the joint distribution of $(Y(0), Y(1))$ conditional on $X = x$.

*First*, we derive the tangent space. The observed data distribution under Assumptions 1 and 3 $(R \perp\!\!\!\perp (Y(0), Y(1)) \mid X$ in $\mathbb{P}_\mathcal{S})$ is given as

$$p(a,x,y,r) = [f(a,y|x)f(x)\pi(x)]^r \cdot [f(a,y|x)f(x)(1 - \pi(x))]^{1-r}$$

$$= f(x) \times \left[\{f_1(y|x)e_1(x)\}^a \{f_0(y|x)(1 - e_1(x))\}^{1-a} \pi(x)\right]^r [f_0(y|x)(1 - \pi(x))]^{1-r}$$

where $f_1(\cdot|x) = \int f(y_0, \cdot|x)dy_0$ and $f_0(\cdot|x) = \int f(\cdot, y_1|x)dy_1$ are the marginal density of $Y(1)$ and $Y(0)$ given $X = x$, respectively. Consider a regular parametric submodel indexed by $\theta$ given as

$$p(a,x,y,r;\theta) = f(x,\theta) \times \left[f_1(y|x,\theta)^a f_0(y|x,\theta)^{1-a}\right]^r \times \left[e_1(x,\theta)^a(1 - e_1(x,\theta))^{1-a}\pi(x,\theta)\right]^r$$

$$\times [f_0(y|x,\theta)(1 - \pi(x,\theta))]^{1-r},$$

which equals $p(a,x,y,r)$ when $\theta = \theta_0$. Also, $f_a(y|x,\theta) = f_a(y|x, r = 1, \theta) = f_a(y|x, r = 0, \theta)$ by Assumption 3. Then, the score function for this submodel is given by

$$s(a,x,y,r;\theta) = \frac{\partial \log p(a,x,y,r;\theta)}{\partial \theta}$$

$$= t(x,\theta) + ra \cdot s_1(y|x,\theta) + (1 - a) \cdot s_0(y|x,\theta)$$

$$+ r\frac{a - e_1(x,\theta)}{e_1(x,\theta)(1 - e_1(x,\theta))}\dot{e}_1(x,\theta) + \frac{r - \pi(x,\theta)}{\pi(x,\theta)(1 - \pi(x,\theta))}\dot{\pi}(x,\theta),$$

where

$$\begin{cases} t(x,\theta) = \partial f(x,\theta)/\partial\theta \\ s_a(y|x,\theta) = \partial \log f_1(y|x,\theta)/\partial\theta \quad \text{for } a = 0, 1 \\ \dot{e}_1(x,\theta) = \partial e_1(x,\theta)/\partial\theta \\ \dot{\pi}(x,\theta) = \partial\pi(x,\theta)/\partial\theta \end{cases}$$

Thus, the tangent space is given as

$$\mathcal{T} = \{t(x) + ras_1(y|x) + (1 - a)s_0(y|x)$$

$$+ r(a - e_1(x)) \cdot b_1(x) + (r - \pi(x)) \cdot b_2(x)\},$$

where $s_a(y|x)$ satisfies $\mathbb{E}[s_a(Y|X) \mid X = x] = \int s_a(y|x)f_a(y|x)dy = 0$ for $a = 0, 1$, $t(x)$ satisfies $\mathbb{E}[t(X)] = \int t(x)f(x)dx = 0$, $b_1(x)$ and $b_2(x)$ are arbitrary square-intergrable measurable function of $x$. In addition, $s_a(y|x) = s_a(y|x, r = 1) = s_a(y|x, r = 0)$ according to $f_a(y|x) = f_a(y|x, r = 1) = f_a(y|x, r = 0)$.

*Second*, we calculate the pathwise derivative of $\tau$. Under the above parametric submodel, the target estimand $\tau = \tau(\theta)$ can be written as

$$\tau(\theta) = \mathbb{E}[Y(1) - Y(0) \mid R = 1]$$

$$= \mathbb{E}_\mathcal{S}[Y(1) - Y(0) \mid R = 1]$$

$$= \frac{\mathbb{E}_\mathcal{S}[RY(1) - RY(0)]}{\mathbb{P}_\mathcal{S}(R = 1)}$$

$$= \frac{\int\int \pi(x,\theta)yf_1(y|x,\theta)f(x,\theta)dydx}{\int \pi(x,\theta)f(x,\theta)dx} - \frac{\int\int \pi(x,\theta)yf_0(y|x,\theta)f(x,\theta)dydx}{\int \pi(x,\theta)f(x,\theta)dx}.$$

By calculation, the pathwise derivative of $\tau(\theta)$ at $\theta = \theta_0$ is given as

$$\frac{\partial \tau(\theta)}{\partial \theta}\bigg|_{\theta=\theta_0} = \frac{\mathbb{E}_{\mathcal{S}}\Big[\pi(X) \cdot \mathbb{E}_{\mathcal{S}}\{Y(1) \cdot s_1(Y(1)|X)|X\}\Big]}{q}$$

$$- \frac{\mathbb{E}_{\mathcal{S}}\Big[\pi(X) \cdot \mathbb{E}_{\mathcal{S}}\{Y(0) \cdot s_0(Y(0)|X)|X\}\Big]}{q}$$

$$+ \frac{\mathbb{E}_{\mathcal{S}}\Big[\big\{\pi(X)t(X) + \dot{\pi}(X)\big\} \cdot \big\{m_1(X) - m_0(X) - \tau\big\}\Big]}{q}.$$

*Third*, we construct the efficient influence function of $\tau$. Let

$$\phi = \frac{\pi(X)}{q}\left[\frac{RA\{Y - m_1(X)\}}{e_{\mathcal{S}}(X)} - \frac{(1-A)\{Y - m_0(X)\}}{1 - e_{\mathcal{S}}(X)}\right] + \frac{R}{q}\{m(X) - m_0(X) - \tau\},$$

where $e_{\mathcal{S}}(X) = \mathbb{P}_{\mathcal{S}}(A = 1|X) = \mathbb{P}_{\mathcal{S}}(A = 1|X, R = 1)\mathbb{P}_{\mathcal{S}}(R = 1|X) + \mathbb{P}_{\mathcal{S}}(A = 1|X, R = 0)\mathbb{P}_{\mathcal{S}}(R = 0|X) = e_1(X)\pi(X)$, $1 - e_{\mathcal{S}}(X) = \mathbb{P}_{\mathcal{S}}(A = 0|X) = (1 - e_1(X))\pi(X) + (1 - \pi(X))$.

*Fourth*, we verify that $\tau$ is an influence function of $\tau$. This holds if it satisfies the following equation

$$\frac{\partial \tau(\theta)}{\partial \theta}\bigg|_{\theta=\theta_0} = \mathbb{E}_{\mathcal{S}}[\phi \cdot s(A, X, Y, R; \theta_0)], \tag{A.1}$$

Next, we give a detailed proof of (A.1).

$$\mathbb{E}_{\mathcal{S}}[\phi \cdot s(A, X, Y, R; \theta_0)] = H_1 + H_2 + H_3,$$

where

$$H_1 = \mathbb{E}_{\mathcal{S}}\left[\left\{\frac{\pi(X)}{q}\frac{RA\{Y - m_1(X)\}}{e_{\mathcal{S}}(X)}\right\} \cdot s(A, X, Y, R; \theta_0)\right],$$

$$= \mathbb{E}_{\mathcal{S}}\left[\left\{\frac{\pi(X)}{q}\frac{RA\{Y - m_1(X)\}}{e_{\mathcal{S}}(X)}\right\} \cdot s_1(Y|X)\right]$$

$$= \mathbb{E}_{\mathcal{S}}\left[\mathbb{E}_{\mathcal{S}}\left\{\frac{\pi(X)}{q}\frac{RA\{Y - m_1(X)\}}{e_{\mathcal{S}}(X)} \cdot s_1(Y|X)\Big|X\right\}\right]$$

$$= \mathbb{E}_{\mathcal{S}}\left[\mathbb{E}_{\mathcal{S}}\left\{\frac{\pi(X)^2}{q}\frac{e_1(X)\{Y(1) - m_1(X)\}}{e_{\mathcal{S}}(X)} \cdot s_1(Y(1)|X)\Big|X, G = 1\right\}\right]$$

$$= \mathbb{E}_{\mathcal{S}}\left[\frac{\pi(X)}{q}\mathbb{E}_{\mathcal{S}}\left\{\{Y(1) - m_1(X)\} \cdot s_1(Y(1)|X)\Big|X\right\}\right]$$

$$= \frac{\mathbb{E}_{\mathcal{S}}\Big[\pi(X) \cdot \mathbb{E}_{\mathcal{S}}\{Y(1) \cdot s_1(Y(1)|X)|X\}\Big]}{q}$$

$$= \text{the first term of } \frac{\partial \tau(\theta)}{\partial \theta}\bigg|_{\theta=\theta_0},$$

$$H_2 = \mathbb{E}_{\mathcal{S}}\left[\left\{\frac{\pi(X)}{q}\frac{(1-A)\{Y - m_0(X)\}}{1 - e_{\mathcal{S}}(X)}\right\} \cdot s(A, X, Y, R; \theta_0)\right],$$

$$= \mathbb{E}_{\mathcal{S}}\left[\left\{\frac{\pi(X)}{q}\frac{(1-A)\{Y - m_0(X)\}}{1 - e_{\mathcal{S}}(X)}\right\} \cdot s_0(Y|X)\right]$$

$$= \mathbb{E}_{\mathcal{S}}\left[\frac{\pi(X)}{q}\frac{1 - e_{\mathcal{S}}(X)}{1 - e_{\mathcal{S}}(X)} \cdot \mathbb{E}_{\mathcal{S}}\left\{Y(0) \cdot s_0(Y(0)|X)\Big|X\right\}\right]$$

$$= \frac{\mathbb{E}_{\mathcal{S}}\Big[\pi(X) \cdot \mathbb{E}_{\mathcal{S}}\{Y(0) \cdot s_0(Y(0)|X)|X\}\Big]}{q}$$

$$= \text{the second term of } \frac{\partial \tau(\theta)}{\partial \theta}\bigg|_{\theta=\theta_0},$$

and

$$H_3 = \mathbb{E}_{\mathcal{S}}\left[\left\{\frac{R}{q}\{m_1(X) - m_0(X) - \tau\}\right\} \times s(A, X, Y, R; \theta_0)\right]$$

$$= \mathbb{E}_{\mathcal{S}}\left[\left\{\frac{R}{q}\{m_1(X) - m_0(X) - \tau\}\right\}\left\{t(X) + R\frac{A - e_1(X)}{e_1(X)(1 - e_1(X))}\dot{e}_1(X) + \frac{R - \pi(X)}{\pi(X)(1 - \pi(X))}\dot{\pi}(X)\right\}\right]$$

$$= \mathbb{E}_{\mathcal{S}}\left[\left\{\frac{R}{q}\{m_1(X) - m_0(X) - \tau\}\right\}\left\{t(X) + \frac{A - e_1(X)}{e_1(X)(1 - e_1(X))}\dot{e}_1(X) + \frac{1 - \pi(X)}{\pi(X)(1 - \pi(X))}\dot{\pi}(X)\right\}\right]$$

$$= \mathbb{E}\left[\left\{\frac{\pi(X)}{q}\{m_1(X) - m_0(X) - \tau\}\right\} \times \left\{t(X) + \frac{1}{\pi(X)}\dot{\pi}(X)\right\}\right]$$

$$= \frac{\mathbb{E}_{\mathcal{S}}\left[\left\{\pi(X)t(X) + \dot{\pi}(X)\right\}\left\{m_1(X) - m_0(X) - \tau\right\}\right]}{q}$$

$$= \text{ the third term of } \left.\frac{\partial \tau(\theta)}{\partial \theta}\right|_{\theta = \theta_0}.$$

Thus, equation (A.1) holds.

*Finally*, we show that $\phi$ is efficient influence function by verifying $\phi \in \mathcal{T}$. Let

$$\begin{cases} t(X) = & \frac{\pi(X)}{q}\{m_1(X) - m_0(X) - \tau\} \\ s_1(Y|X) = & \frac{\pi(X)}{q}\frac{(Y - m_1(X))}{e_{\mathcal{S}}(X)} \\ s_0(Y|X) = & \frac{\pi(X)}{q}\frac{(Y - m_0(X))}{1 - e_{\mathcal{S}}(X)} \\ b_2(X) = & \frac{m_1(X) - m_0(X) - \tau}{1 - q}, \end{cases}$$

then $\phi$ can be written as

$$\phi = t(X) + RAs_1(Y|X) + (1 - A)s_0(Y|X) + (R - \pi(X))b_2(X).$$

Clearly, $\int s_a(y|x)f_a(y|x)dy = 0$ for $a = 0, 1$, and $\int t(x)f(x)dx = 0$, which implies that $\phi \in \mathcal{T}$, and thus $\phi$ is the efficient influence function of $\tau$.

$\square$

**Lemma 2.** Under Assumptions 1 and 3, if $||\hat{e}_1(x) - e_1(x)||_2 \cdot ||\hat{m}_a(x) - m_a(x)||_2 = o_{\mathbb{P}}(n^{-1/2})$ and $||\hat{\pi}(x) - \pi(x)||_2 \cdot ||\hat{m}_a(x) - m_a(x)||_2 = o_{\mathbb{P}}(n^{-1/2})$ for all $x \in \mathcal{X}$ and $a \in \{0, 1\}$, then $\hat{\tau}_{\mathcal{S}}$ satisfies $\sqrt{N_{\mathcal{E}} + N_{\mathcal{S}}}(\hat{\tau}_{\mathcal{S}} - \tau) \xrightarrow{d} \mathcal{N}(0, \sigma^2)$, where $\sigma^2$ is the semiparametric efficiency bound of $\tau$, and $\xrightarrow{d}$ means convergence in distribution.

**Proof of Lemma 2.** Let $Z = (X, A, R, Y)$, and denote

$$\tilde{\phi}(Z; m_0, m_1, \pi, e_1) = \frac{\pi(X)}{q}\left[\frac{RA\{Y - m_1(X)\}}{e_{\mathcal{S}}(X)} - \frac{(1 - A)\{Y - m_0(X)\}}{1 - e_{\mathcal{S}}(X)}\right]$$
$$+ \frac{R}{q}\{m_1(X) - m_0(X)\}$$

as the non-centralized efficient influence functions of $\tau$, where $m_0, m_1, \pi, e_1$ are the nuisance parameters $m_0(x), m_1(x), \pi(x), e_1(x)$, respectively. We introduce them for ease of presentation.

Then $\hat{\tau}$ can be written as

$$\hat{\tau} = \frac{1}{N_{\mathcal{E}} + N_{\mathcal{S}}}\sum_{i \in \mathcal{E} \cup \mathcal{S}}[\tilde{\phi}_1(Z; \hat{m}_0, \hat{m}_1, \hat{\pi}, \hat{e}_1)],$$

where $(\hat{m}_0, \hat{m}_1, \hat{\pi}, \hat{e}_1)$ are estimates of $(m_0, m_1, \pi, e_1)$.

Let $n = N_{\mathcal{E}} + N_{\mathcal{S}}$, we decompose $\hat{\tau} - \tau$ as

$$\hat{\tau} - \tau = U_{1n} + U_{2n},$$

where

$$U_{1n} = \frac{1}{n} \sum_{i \in \mathcal{E} \cup \mathcal{S}} [\tilde{\phi}(Z_i; \mu_0, \mu_1, \pi, e_1) - \tau],$$

$$U_{2n} = \frac{1}{n} \sum_{i \in \mathcal{E} \cup \mathcal{S}} [\tilde{\phi}(Z_i; \hat{\mu}_0, \hat{\mu}_1, \hat{\pi}, \hat{e}_1) - \tilde{\phi}(Z_i; \mu_0, \mu_1, \pi, e_1)].$$

Note that $U_{1n}$ is a sum of $n$ independent variables with zero means, and its variance equals $\mathbb{V}^*/n$, where $\mathbb{V}^*$ is the semiparametric efficiency bound. By the central limit theorem, we have

$$\sqrt{n} U_{1n} \xrightarrow{d} N(0, \mathbb{V}^*),$$

where $\xrightarrow{d}$ denotes convergence in distribution. Thus, it suffices to show that $U_{2n} = o_{\mathbb{P}}(n^{-1/2})$. $U_{2n}$ can be be further decomposed as

$$U_{2n} = U_{2n} - \mathbb{E}_{\mathcal{S}}[U_{2n}] + \mathbb{E}_{\mathcal{S}}[U_{2n}].$$

By a Taylor expansion for $\mathbb{E}[U_{2n}]$ yields that

$$\begin{aligned}
\mathbb{E}_{\mathcal{S}}[U_{2n}] &= \mathbb{E}_{\mathcal{S}}[\tilde{\phi}(Z; \hat{m}_0, \hat{m}_1, \hat{\pi}, \hat{e}_1) - \tilde{\phi}(Z; m_0, m_1, \pi, e_1)] \\
&= \partial_{[\hat{m}_0 - m_0, \hat{m}_1 - m_1, \hat{\pi} - \pi, \hat{e}_1 - e_1]} \mathbb{E}_{\mathcal{S}}[\tilde{\phi}(Z; m_0, m_1, \pi, e_1)] \\
&\quad + \frac{1}{2} \partial^2_{[\hat{m}_0 - m_0, \hat{m}_1 - m_1, \hat{\pi} - \pi, \hat{e}_1 - e_1]} \mathbb{E}_{\mathcal{S}}[\tilde{\phi}(Z; m_0, m_1, \pi, e_1)] \\
&\quad + \cdots
\end{aligned}$$

The first-order term

$$\begin{aligned}
&\partial_{[\hat{m}_0 - m_0, \hat{m}_1 - m_1, \hat{\pi} - \pi, \hat{e}_1 - e_1]} \mathbb{E}_{\mathcal{S}}[\tilde{\phi}(Z; m_0, m_1, \pi, e_1)] \\
&= \mathbb{E}_{\mathcal{S}}\left[ -\frac{1}{q} \left\{ R - \frac{\pi(X)(1-A)}{1 - e_{\mathcal{S}}(X)} \right\} (\hat{m}_0(X) - m_0(X)) \right] \\
&\quad + \mathbb{E}_{\mathcal{S}}\left[ \frac{1}{q} \left\{ R - \frac{\pi(X)RA}{e_{\mathcal{S}}(X)} \right\} (\hat{m}_1(X) - m_1(X)) \right] \\
&\quad + \mathbb{E}_{\mathcal{S}}\left[ \frac{1}{q} \left\{ \frac{RA\{Y - m_1(X)\}}{e_{\mathcal{S}}(X)} + \frac{(1-A)\{Y - m_0(X)\}}{1 - e_{\mathcal{S}}(X)} \right\} (\hat{\pi}(X) - \pi(X)) \right] \\
&\quad - \mathbb{E}_{\mathcal{S}}\left[ \frac{\pi(X)}{q} \left\{ \frac{RA\{Y - m_1(X)\}}{e_{\mathcal{S}}(X)^2} + \frac{(1-A)\{Y - m_0(X)\}}{\{1 - e_{\mathcal{S}}(X)\}^2} \right\} \times e_1(X)(\hat{\pi}(X) - \pi(X)) \right] \\
&\quad - \mathbb{E}_{\mathcal{S}}\left[ \frac{\pi(X)}{q} \left\{ \frac{RA\{Y - m_1(X)\}}{e_{\mathcal{S}}(X)^2} + \frac{(1-A)\{Y - m_0(X)\}}{\{1 - e_{\mathcal{S}}(X)\}^2} \right\} \times \pi(X)(\hat{e}_1(X) - e_1(X)) \right] \\
&= 0.
\end{aligned}$$

where the last equation follows from $\mathbb{E}_{\mathcal{S}}[A|X] = e_{\mathcal{S}}(X)$, $\mathbb{E}_{\mathcal{S}}[\pi(X)A \mid X] = \pi(X)e_1(X)$, $\mathbb{E}[RA(Y - m_1(X))|X] = 0$, and $\mathbb{E}[(1-A)(Y - m_0(X))|X] = 0$.

For the second-order term, we get

$$\partial^2_{[\hat{m}_0-m_0,\hat{m}_1-m_1,\hat{\pi}-\pi,\hat{e}_1-e_1]}\mathbb{E}_{\mathcal{S}}[\tilde{\phi}(Z;m_0,m_1,\pi,e_1)]$$

$$= \mathbb{E}_{\mathcal{S}}\left[\frac{1}{q}\frac{(1-A)}{1-e_{\mathcal{S}}(X)}(\hat{m}_0(X)-m_0(X))(\hat{\pi}(X)-\pi(X))\right]$$

$$+ \mathbb{E}_{\mathcal{S}}\left[\frac{1}{q}\frac{\pi(X)(1-A)}{\{1-e_{\mathcal{S}}(X)\}^2}e_1(X)(\hat{m}_0(X)-m_0(X))(\hat{\pi}(X)-\pi(X))\right]$$

$$+ \mathbb{E}_{\mathcal{S}}\left[\frac{1}{q}\frac{\pi(X)(1-A)}{\{1-e_{\mathcal{S}}(X)\}^2}\pi(X)(\hat{m}_0(X)-m_0(X))(\hat{e}_1(X)-e_1(X))\right]$$

$$- \mathbb{E}_{\mathcal{S}}\left[\frac{1}{q}\frac{RA}{e_{\mathcal{S}}(X)}(\hat{m}_1(X)-m_1(X))(\hat{\pi}(X)-\pi(X))\right]$$

$$+ \mathbb{E}_{\mathcal{S}}\left[\frac{1}{q}\frac{\pi(X)RA}{e_{\mathcal{S}}(X)^2}e_1(X)(\hat{m}_1(X)-m_1(X))(\hat{\pi}(X)-\pi(X))\right]$$

$$+ \mathbb{E}_{\mathcal{S}}\left[\frac{1}{q}\frac{\pi(X)RA}{e_{\mathcal{S}}(X)^2}\pi(X)(\hat{m}_1(X)-m_1(X))(\hat{e}_1(X)-e_1(X))\right]$$

$$+ \mathbb{E}_{\mathcal{S}}\left[\frac{1}{q}\frac{(1-A)}{e_{\mathcal{S}}(X)}(\hat{\pi}(X)-\pi(X))(\hat{m}_0(X)-m_0(X))\right]$$

$$- \mathbb{E}_{\mathcal{S}}\left[\frac{1}{q}\frac{RA}{e_{\mathcal{S}}(X)}(\hat{\pi}(X)-\pi(X))(\hat{m}_1(X)-m_1(X))\right]$$

$$+ \mathbb{E}_{\mathcal{S}}\left[\frac{1}{q}\frac{(1-A)}{\{1-e_{\mathcal{S}}(X)\}^2}e_1(X)(\hat{\pi}(X)-\pi(X))(\hat{m}_0(X)-m_0(X))\right]$$

$$+ \mathbb{E}_{\mathcal{S}}\left[\frac{1}{q}\frac{RA}{e_{\mathcal{S}}(X)^2}e_1(X)(\hat{\pi}(X)-\pi(X))(\hat{m}_1(X)-m_1(X))\right]$$

$$+ \mathbb{E}_{\mathcal{S}}\left[\frac{1}{q}\frac{(1-A)}{\{1-e_{\mathcal{S}}(X)\}^2}\pi(X)(\hat{e}_1(X)-e_1(X))(\hat{m}_0(X)-m_0(X))\right]$$

$$+ \mathbb{E}_{\mathcal{S}}\left[\frac{\pi(X)}{q}\frac{A}{e_{\mathcal{S}}(X)^2}\pi(X)(\hat{e}_1(X)-e_1(X))(\hat{m}_1(X)-m_1(X))\right]$$

$$= O_{\mathbb{P}}\Big(||\hat{e}_1(X)-e_1(X)||_2\cdot(||\hat{m}_1(X)-m_1(X)||_2+||\hat{m}_0(X)-m_0(X)||_2)$$

$$+ ||\hat{\pi}(X)-\pi(X)||_2\cdot(||\hat{m}_1(X)-m_1(X)||_2+||\hat{m}_0(X)-m_0(X)||_2)\Big)$$

$$= o_{\mathbb{P}}(n^{-1/2}),$$

All higher-order terms can be shown to be dominated by the second-order term. Therefore, $\mathbb{E}_{\mathcal{S}}[U_{2n}] = o_{\mathbb{P}}(n^{-1/2})$. In addition, we get that $U_{2n} - \mathbb{E}_{\mathcal{S}}[U_{2n}] = o_{\mathbb{P}}(n^{-1/2})$ by calculating $\text{Var}\{\sqrt{n}(U_{2n} - \mathbb{E}_{\mathcal{S}}[U_{2n}])\} = o_{\mathbb{P}}(1)$. This proves the conclusion.

$\square$

**Theorem 1 (Bias-Variance Analysis).** Under Assumption 1 only, if $||\hat{e}_1(x) - e_1(x)||_2 \cdot ||\hat{m}_a(x) - m_a(x)||_2 = o_{\mathbb{P}}(n^{-1/2})$ and $||\hat{\pi}(x) - \pi(x)||_2 \cdot ||\hat{m}_a(x) - m_a(x)||_2 = o_{\mathbb{P}}(n^{-1/2})$ for all $x \in \mathcal{X}$ and $a \in \{0, 1\}$, then $\hat{\tau}_{\mathcal{S}}$ satisfies $\sqrt{N_{\mathcal{E}} + N_{\mathcal{S}}}\{\hat{\tau}_{\mathcal{S}} - \tau - \text{bias}(\hat{\tau}_{\mathcal{S}})\} \xrightarrow{d} \mathcal{N}(0, \sigma^2)$, where $\sigma^2 = \text{Var}(\phi)$, $\phi$ is defined in Proposition 1, and $\text{bias}(\hat{\tau}_{\mathcal{S}}) = \mathbb{E}_{\mathcal{S}}\left[\frac{R}{q}\{m_0(X) - \mu_0(X)\}\right]$.

**Proof of Theorem 1**. By the proof of Proposition 2, if $||\hat{e}_1(x) - e_1(x)||_2 \cdot ||\hat{m}_a(x) - m_a(x)||_2 = o_\mathbb{P}(n^{-1/2})$ and $||\hat{\pi}(x) - \pi(x)||_2 \cdot ||\hat{m}_a(x) - m_a(x)||_2 = o_\mathbb{P}(n^{-1/2})$, we have

$$\hat{\tau}_\mathcal{S} = \frac{1}{N_\mathcal{E} + N_\mathcal{S}} \sum_{i \in \mathcal{E} \cup \mathcal{S}} \frac{\pi(X_i)}{q} \left[ \frac{R_i A_i (Y_i - m_1(X_i))}{e_\mathcal{S}(X_i)} - \frac{(1 - A_i)(Y_i - m_0(X_i))}{1 - e_\mathcal{S}(X_i)} \right]$$

$$+ \frac{1}{N_\mathcal{E} + N_\mathcal{S}} \sum_{i \in \mathcal{E} \cup \mathcal{S}} \frac{R_i}{q} \{m_1(X_i) - m_0(X_i)\} + o_\mathbb{P}(n^{-1/2}).$$

Note that by definition, $\mu_1(X) = m_1(X)$ but $\mu_0(X) \neq m_0(X)$. Then, under Assumption 1,

$$\tau = \mathbb{E}\left[ \frac{R}{q} \{\mu_1(X) - \mu_0(X)\} \right] = \mathbb{E}\left[ \frac{R}{q} \{m_1(X) - \mu_0(X)\} \right] = \mathbb{E}_\mathcal{S}\left[ \frac{R}{q} \{m_1(X) - \mu_0(X)\} \right].$$

The bias of $\hat{\tau}_\mathcal{S}$ is

$$\mathbb{E}_\mathcal{S}\left[ \frac{\pi(X_i)}{q} \left\{ \frac{R_i A_i (Y_i - m_1(X_i))}{e_\mathcal{S}(X_i)} - \frac{(1 - A_i)(Y_i - m_0(X_i))}{1 - e_\mathcal{S}(X_i)} \right\} + \frac{R_i}{q} \{m_1(X_i) - m_0(X_i)\} \right] - \tau$$

$$= \mathbb{E}_\mathcal{S}\left[ \frac{R}{q} \{m_1(X) - m_0(X)\} \right] - \mathbb{E}_\mathcal{S}\left[ \frac{R}{q} \{m_1(X) - \mu_0(X)\} \right]$$

$$= \mathbb{E}_\mathcal{S}\left[ \frac{R}{q} \{\mu_0(X) - m_0(X)\} \right].$$

Therefore,

$$\sqrt{N_\mathcal{E} + N_\mathcal{S}} \{\hat{\tau}_\mathcal{S} - \tau - \mathrm{bias}(\hat{\tau}_\mathcal{S})\}$$

$$= \frac{1}{\sqrt{N_\mathcal{E} + N_\mathcal{S}}} \sum_{i \in \mathcal{E} \cup \mathcal{S}} \frac{\pi(X_i)}{q} \left[ \frac{R_i A_i (Y_i - m_1(X_i))}{e_\mathcal{S}(X_i)} - \frac{(1 - A_i)(Y_i - m_0(X_i))}{1 - e_\mathcal{S}(X_i)} \right]$$

$$+ \frac{1}{\sqrt{N_\mathcal{E} + N_\mathcal{S}}} \sum_{i \in \mathcal{E} \cup \mathcal{S}} \left[ \frac{R_i}{q} \{m_1(X_i) - m_0(X_i)\} - \mathbb{E}_\mathcal{S}[\frac{R_i}{q} \{m_1(X_i) - m_0(X_i)\}] \right]$$

$$+ o_\mathbb{P}(1).$$

This implies the conclusion by central limit theorem.

$\square$

# B  Additional Experimental Results

## B.1  Sensitivity Analysis on Control-n

To further demonstrate the robustness of the proposed approach $\hat{\tau}_{\text{if}}$. We consider different control group sizes in RCT, in the simulation study, we consider control group sizes of $N_c^s \in \{70, 80, 90\}$ for "Linear" and "Nonlinear", while for the real-data application, we examine a range of $N_c^s \in \{70, 75, 85\}$. The results is shown in the Figure A1, A2 and A3.

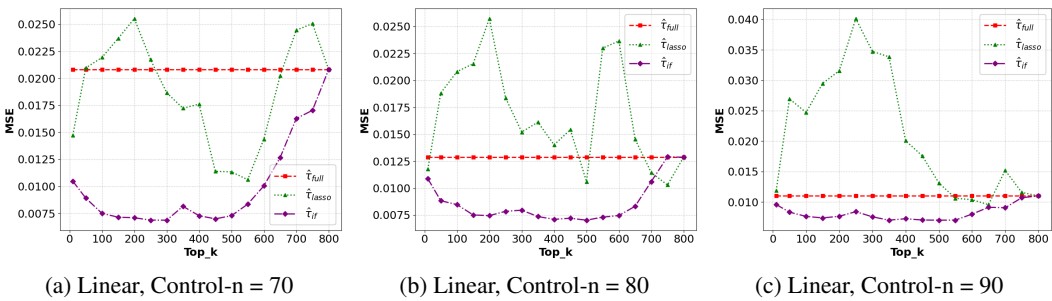

(a) Linear, Control-n = 70       (b) Linear, Control-n = 80       (c) Linear, Control-n = 90

Figure A1: Comparison of the performance of different approaches in "Linear" at different $N_c^s$.

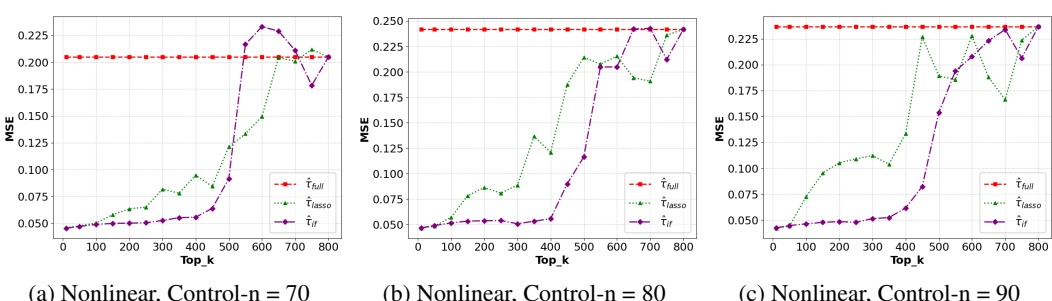

(a) Nonlinear, Control-n = 70     (b) Nonlinear, Control-n = 80     (c) Nonlinear, Control-n = 90

Figure A2: Comparison of the performance of different approaches in "Nonlinear" at different $N_c^s$.

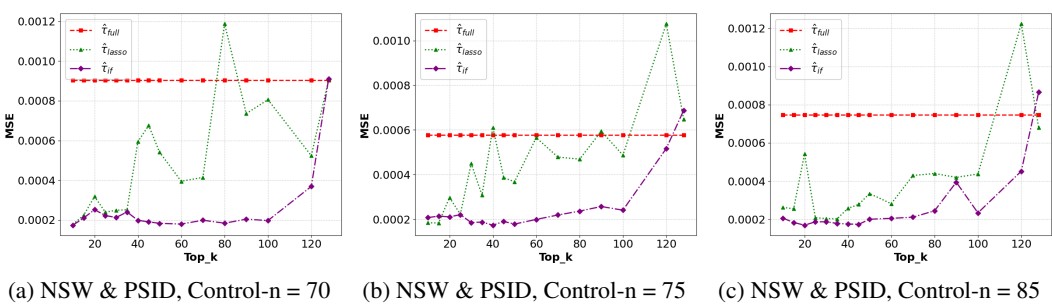

(a) NSW & PSID, Control-n = 70    (b) NSW & PSID, Control-n = 75    (c) NSW & PSID, Control-n = 85

Figure A3: Comparison of the performance of different approaches in real-data at different $N_c^s$.

## B.2  Sensitivity Analysis on Covariate Shift

To further demonstrate the robustness of the proposed approach $\hat{\tau}_{\text{if}}$. We consider different degrees of covariate shift $\mu_2 = \{0.2, 0.3, 0.4\}$ in "Nonlinear" as shown in Figure A4.

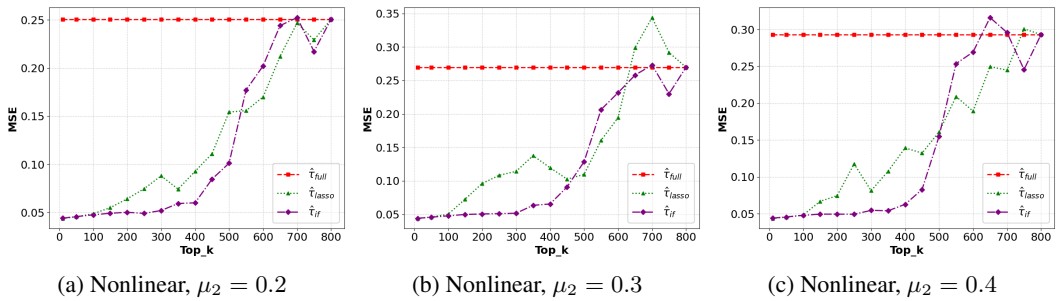

| (a) Nonlinear, $\mu_2 = 0.2$ | (b) Nonlinear, $\mu_2 = 0.3$ | (c) Nonlinear, $\mu_2 = 0.4$ |

Figure A4: Comparison of the performance of different approaches in "Nonlinear" at different $\mu_2$.

## C   Implementation Details

For real-world data, we train the outcome model of the RCT data or external control through Multi-Layer Perceptron (MLP). In the estimation phase, the outcome regression $\hat{\mu}_0$, $\hat{\mu}_1$, and $\hat{m}_0$, $\hat{\mu}_{0,\mathcal{O}}$ are modeled using a Multi-Layer Perceptron (MLP), while both the propensity score and selection score are estimated via logistic regression. As shown in Table A1, it presents the hyperparameter space for outcome regression $\hat{\mu}_0$, $\hat{\mu}_1$, and $\hat{m}_0$, $\hat{\mu}_{0,\mathcal{O}}$ in both simulated and real-world datasets.

Table A1: Implementation Details

| Hyperparameter | $\hat{\mu}_0$ | $\hat{\mu}_1$ | $\hat{m}_0$ | $\hat{\mu}_{0,\mathcal{O}}$ |
|---|---|---|---|---|
| Learning rate | 0.0005 | 0.0005 | 0.0005 | 0.0005 |
| Batch size | 32 | 32 | 32 | 32 |
| Architecture | 1 hidden layers [16] | 2 hidden layers [16,8] | 2 hidden layers [16,8] | 2 hidden layers [16,8] |
| Optimizer | Adam | Adam | Adam | Adam |
| Early stopping patience | 20 | 20 | 20 | 20 |
| Activation function (all layers) | ReLU | ReLU | ReLU | ReLU |

