# OpenReview forum: "Adaptive Data-Borrowing for Improving Treatment Effect Estimation using External Controls"
_NeurIPS.cc/2025/Conference — NeurIPS 2025 poster_

### Official Review · Reviewer_uy6o · 2025-07-02

**Clarity:** 3
**Significance:** 3
**Originality:** 2
**Rating:** 4
**Confidence:** 3

**Summary:**

This paper studies how to use external control data to improve inferential efficiency in estimating treatment effects for randomized controlled trials (RCTs), especially those with small sample sizes. The main difficulty arises from the individual heterogeneity; that is, the potential outcomes of external control units may be different from those of RCT control units. To address this issue, the authors proposed an influence-based adaptive sample borrowing approach that quantifies the comparability of each unit in the external control and adaptively adds external units to enhance the efficiency of estimating treatment effects. Specifically, the method calculates an influence score for each external unit based on the change of a pre-specified loss function when the external unit is added to learn the model parameters. Then, the authors derived a semiparametric estimator for the average treatment effect (a doubly robust estimator), analyzed its bias, and proved asymptotic normality of the estimation error.

**Questions:**

1. I am confused about Proposition 1. What is the "efficient influence function"? It seems [1] also provides a formula for the efficient influence function under the same assumptions if one replaces $P$ with $P_S$. Is the formula in Proposition 1 different from that in [1]? If so, why are they different? In addition, what is the parameter $q$ in Proposition 1?

2. In Section 5, when learning the model $\hat{\theta}_z$ parameters by the empirical risk minimizer, there is a regularizer term; however, when using the influence function to approximate the model change $\hat{\theta}$$\epsilon, z$, there is no regularizer term. Could the authors explain this discrepancy?

3. In the experiment setup for the NSW data, could the authors explain how the unobserved outcomes are generated? Every unit has a missing potential outcome. I assume that one needs all potential outcomes to calculate the MSE?

4. In Proposition 2 and Theorem 1, should $(N_E + N_S) (\hat{\tau}_S - \tau) \rightarrow N(0, \sigma^2)$ be $\sqrt{N_E + N_S} (\hat{\tau}_S - \tau) \rightarrow N(0, \sigma^2)$?

References:
[1] Chenyin Gao, Shu Yang, Mingyang Shan, Wenyu YE, Ilya Lipkovich, and Douglas Faries. Improving randomized controlled trial analysis via data-adaptive borrowing. Biometrika, page asae069, 2024.

**Ethical Concerns:**

["NO or VERY MINOR ethics concerns only"]

**Final Justification:**

The paper studies a well-motivated research problem and provides new ideas for borrowing external data to improve causal estimation. Although I found the theoretical analysis somewhat limited, I think the paper is overall interesting. I also appreciate the authors' detailed responses to my questions during the rebuttal. I increased my score. A minor suggestion for the authors is to provide a clearer explanation of the "optimal borrowed set" in a future version.

**Limitations:**

yes

**Quality:**

2

**Strengths And Weaknesses:**

### Strengths:
1. The research problem is well-motivated. Due to small sample sizes of RCTs, one wants to combine external control data to improve the efficiency of estimating treatment effects. However, most existing works rely on a potentially implausible assumption that external control and RCT control units are exchangeable. The authors aim to drop this exchangeability assumption. The introduction is well written.

2. The authors proposed a new influence-based method to adaptively borrow external control units. This new method can handle external outliers effectively.

### Weaknesses:
1. Compared with [1], it seems that the main difference appears to be how to select the external control units. The semiparametric estimator and the asymptotic normality of the estimation error seem similar to those in [1]. The authors could consider providing a more detailed comparison with [1].

2. While the authors presented numerical evidence showing improvement over [1] when the external control data contain outliers, the authors could consider providing some theoretical results that specify the conditions under which their method has a smaller MSE.

3. The authors should provide more details on how to estimate the parameters in their estimator, such as $\hat{\pi}$ (I understand that the authors mentioned $\hat{m}$ is estimated using standard machine learning methods, but I don't think they discussed $\hat{\pi}$). In addition, I think the propensity score of the RCT units (i.e., $\hat{e}_1$) is known by the RCT design.

4. I am confused about Section 6.2 -- find the "optimal" borrowed set. Could the authors elaborate on how the borrowed set S that minimizes the MSE is selected? Do you greedily add external units based on their influence scores? If so, why does this method find the optimal S?

5. The authors may also compare their method with other classical methods, such as matching.

6. I think the authors should mention that they assume that the unit data are i.i.d. samples. This assumption is used in their proof of asymptotic normality of the estimation error.

References:
[1] Chenyin Gao, Shu Yang, Mingyang Shan, Wenyu YE, Ilya Lipkovich, and Douglas Faries. Improving randomized controlled trial analysis via data-adaptive borrowing. Biometrika, page asae069, 2024.

---

> ### Author Rebuttal · Authors · 2025-07-29
>
> **We thank you for your efforts in evaluating our work. Below, we hope that our clarification addresses your concerns and sincerely invite you to kindly re-evaluate our work in light of our explanation.**
>
> > **W1:** Compared with [1], it seems that the main difference appears to be in how to select the external control units. The semiparametric estimator seems similar to those in [1]. The authors could consider providing a more detailed comparison with [1].
>
> **Response to W1:** Thanks for your insightful comments. We agree that both studies share common ground in the semiparametric theory. However, **we would like to clarify that our primary contributions are not in the estimation itself, but in developing a novel approach for selecting external comparable samples and determining the optimal selection set.**
>
> - In Section 4, we provided a comparison between our influence-based method and the lasso-based approach introduced in [1], summarized in the table below.
> - In Sections 5--6, we developed the influence-based method and the optimal borrowed set selection method.
>
> |   |model-free for external controls | robust to outliers  |
> |---|------|--|
> | the adaptive lasso-based method[1] | ×   | ×    |
> | the influence-based method      | √     | √    |
>
> **In addition, in lines 61--68 (the end of Introduction), we did not claim any contribution to semiparametric efficiency. We only claimed the following main contributions:**
> - We reveal the limitations of existing approaches for estimating treatment effects by combining RCT data with external controls.
> - We propose an influence-based sample borrowing approach, which can effectively quantify the comparability of each sample in the external controls.
> -  We develop a data-driven approach to select the optimal subset of external control samples based on the MSE of the proposed estimator.
> - We conduct extensive experiments on both simulated and real-world datasets, demonstrating that the proposed approach outperforms the existing baseline approaches.
>
> > **W2:** The authors could consider providing some theoretical results that specify the conditions under which their method has a smaller MSE.
>
> **Response to W2:**  Thanks for your suggestions. **We are not aware of any settings where the lasso-based method is better than ours. If you have further comments on this, we welcome further discussion.**
>
> **In addition, we would like to clarify that our method has two key merits compared with [1]: Better comparability and insensitivity to outliers, please refer to Section 4 for more details. These two salient merits stem from the fact that ''our method is model-free for outcome model in the external controls.'**' In contrast, the lasso-based method [1] is model-dependent and relies on the outcome model of the external control data, and the simple difference between the two outcome models cannot fully capture the comparability of the external data, as discussed in Section 4.
>
> > **W3:** How to estimate $\hat \pi$. In addition, I think $\hat e_1$ is known by the RCT design.
>
> **Response to W3:** We appreciate your suggestions. In our experiments, we adopt a logistic regression model for estimating $\hat \pi$. **We acknowledge that $\hat{e}_1(X)$ is sometimes known in the context of RCTs. However, we would like to clarify that our proposed method is directly applicable in such cases (as all theoretical results also hold).** We will make this point clearer in the revised version.
>
> > **W4:** I am confused about Section 6.2 -- find the "optimal" borrowed set. Could the authors elaborate on it?
>
> **Response to W4:** Thanks for your comments, and we apologize for the lack of clarity on selecting the "optimal" borrowed set. Below, we provide a detailed description.
> **The proposed method for selecting the "optimal" borrowed set consists of two steps:**
> - Step 1:  Based on the ranking of influence scores, we construct a series of nested subsets of external controls, where each subset comprises the top-K most comparable samples (i.e., the top-K samples with the smallest influence scores), see lines 183--185 of the manuscript.
> - Step 2: Find the optimal $K$ that minimizes the MSE based on Theorem 1, see Section 6.2 of the manuscript.
>
> **From Figure 2 of the manuscript, we can see that the MSE curve will show a trend of first decreasing and then increasing. Then the lowest point of the curve corresponds to the optimal $K$ with the smallest empirical MSE.**
>
> > **W5:** The authors may also compare their method with other classical methods, such as matching.
>
> **Response to W5:** Thank you for raising this interesting question. To the best of our knowledge, there is currently no matching method specifically designed for adaptive selecting external samples.
>
> **We assume the reviewer may be suggesting a strategy in which external control samples that are close to at least one sample in the RCT are selected, while those far from the RCT data are excluded. If this is the case, we would like to clarify that such a matching approach cannot effectively distinguish between 'comparable' and 'non-comparable' samples.**
>
> **The reason is as follows: typical matching methods rely on covariate distance, based on the assumption that
> ''units with similar $x$ also have similar potential outcomes.''
> However, such methods ignore outcome information and may inadvertently match 'non-comparable' samples--particularly when some external control units exhibit conditional outcome patterns (i.e., $E[Y|X]$) that differ from those in the RCT.**
>
>  For example, from Figure 2 in the manuscript,  we could see that all the range of $x$ in the external controls and RCTs are the same. If we conduct matching based on $x$, then we will treat all external controls as "comparable" samples. Clearly, it is not right.
>
> > **W6:** I think the authors should mention that they assume that the unit data are i.i.d. samples.
>
> **Response to W6:** We apologize for the lack of details. We will revise the manuscript to present it clearly.
>
> > **Q1:** I am confused about Proposition 1. It seems [1] also provides a formula for the efficient influence function under the same assumptions if one replaces $P$ with $P_S$. In addition, what is the parameter $q$ in Proposition 1?
>
> **Response to Q1:**  Thank you for raising this issue. The efficient influence function is similar to that of [1] if $P$ is replaced by $P_{S}$, and $q=P_{S}(R=1)$ represents the proportion of RCT data (i.e., R=1) within the combined population $P_{S}$. We will clarify it in the revised version. Thanks again.
>
> **In addition, as our response to W1, our primary contributions are not in the estimation itself, but in developing a novel approach for selecting external comparable samples and determining the optimal selection set
> Therefore, Proposition 1 is just the beginning of our estimation method.**
>
> > **Q2:** In Section 5, when using the influence function to approximate the $\hat \theta_{\epsilon,z}$, there is no regularizer term.
>
> **Response to Q2:** Thanks for raising this point. **We would like to clarify that, theoretically, the influence function approximation of the model change $\hat \theta_{\epsilon,z}$ does not require a regularizer.** Below, we provide the formal derivation:
> \\\[\
>     \\hat \\theta\_{\\epsilon,z}\\triangleq
>     \\arg\\min\_{\\theta}\\frac{1}{N\_{C}}\\sum\_{Z_i\\in C}L(Z_i,\\theta)+\\epsilon
>     L (z;\\theta)+\\frac{\\lambda}{2}\\Vert\\theta\\Vert_2\^2\
>     \\\]\
> By definition,\
> \\\[\
>     0=\\frac{1}{N\_{C}}\\sum\_{Z_i\\in C}\\frac{\\partial
>       L (Z_i,\\hat{\\theta}\_{\\epsilon,z})}{\\partial
>         \\theta}+\\epsilon\\cdot\\frac{\\partial
>           L (z,\\hat{\\theta}\_{\\epsilon,z})}{\\partial
>             \\theta}+\\frac{\\lambda}{2}\\hat{\\theta}\_{\\epsilon,z}.\
>     \\\]\
> By Taylor expansion with respect to \$\\theta\$ at \$\\hat{\\theta}\$,
> we have that,\
> \\\[\
>     0 \\approx
>     \\frac{1}{N\_{C}}\\sum\_{Z_i\\in C}\\frac{\\partial
>       L (Z_i,\\hat{\\theta}\_{\\epsilon,z})}{\\partial
>         \\theta}+\\epsilon\\cdot\\frac{\\partial
>           L (z,\\hat{\\theta}\_{\\epsilon,z})}{\\partial
>             \\theta}+\\frac{\\lambda}{2}\\hat{\\theta}\_{\\epsilon,z}+(\\frac{1}{N\_{C}}\\sum\_{Z_i\\in C}\\frac{\\partial\^2
>               L (Z_i,\\hat{\\theta})}{\\partial
>                 \\theta\^2}+\\epsilon\\cdot\\frac{\\partial\^2
>                   L (z,\\hat{\\theta})}{\\partial
>                     \\theta\^2}+\\frac{\\lambda}{2})\\cdot(\\hat{\\theta}\_{\\epsilon,z}-\\hat{\\theta}).\
>     \\\]\
> Then we have,\
> \\\[\
>     \\hat{\\theta}\_{\\epsilon,z}-\\hat{\\theta}\\approx
>     \[\\frac{1}{N\_{C}}\\sum\_{Z_i\\in C}\\frac{\\partial\^2
>       L (Z_i,\\hat{\\theta})}{\\partial
>         \\theta\^2}+\\epsilon\\cdot\\frac{\\partial\^2
>           L (z,\\hat{\\theta})}{\\partial
>             \\theta\^2}+\\frac{\\lambda}{2}\]\^{-1}\\cdot \\epsilon
>     \\cdot\\frac{\\partial L (z,\\hat{\\theta})}{\\partial \\theta}.\
>     \\\]
> Thus, when $\lambda\to 0$,\
> \\\[\
>     \\left.\\frac{d \\hat{\\theta}\_{\\epsilon,z}}{d \\epsilon}
>     \\right\|\_{\\epsilon = 0}\\approx
>     \[\\frac{1}{N\_{C}}\\sum\_{Z_i\\in C}\\frac{\\partial\^2
>       L (Z_i,\\hat{\\theta})}{\\partial \\theta\^2}\]\^{-1}\\cdot
>     \\frac{\\partial L (z,\\hat{\\theta})}{\\partial \\theta}.\
>     \\\]
>
> > **Q3:** For the NSW data, could the authors explain how the unobserved outcomes are generated?
>
> **Response to Q3:** Thanks for your comments. **We would like to clarify that we did not generate the unobserved outcomes for the NSW data. Below, we give a detailed description of how to calculate the MSE with the observed data only.**
> - Calculate $\hat\tau_{S}$ and $\hat\tau_{aipw}$;
> - Calculate $\textup{bias}(\hat \tau_{S}) = \hat \tau_{S}-\hat \tau_{aipw}$ and $\textup{var}(\hat \tau_{S}) = \textup{var}(\hat \phi)$
> - Calculate MSE: $\textup{MSE}(\hat \tau_S) = \textup{bias}^2(\hat \tau_S) + \textup{var}(\hat \tau_S)$.
>
> **Response to Q4:** Thanks for pointing this out. Yes, we will revise it.

---

> > ### Comment · Reviewer_uy6o · 2025-08-06
> >
> > I thank the authors for their detailed response to my questions and concerns.
> >
> > W1: It seems that Theorem 1 (without Assumption 3: exchangeability for borrowed external controls) is a shifted version of Proposition 2 (under Assumption 3). Without Assumption 3, $E[\hat{\tau}_S] - \tau$ may not be zero, but one can subtract this term and get the asymptotical normality. I feel this result seems standard. It would also be helpful if the authors could provide a theoretical characterization of the MSE of their method.
> >
> > W4: For the optimal borrowed set, what's the definition of "optimal"? In Section 5, only the loss of a single external control sample is defined. For a set z, should we treat the $\hat{\theta}_{+z}$ as the model parameter after adding the whole set z? If so, how do the authors prove the optimality of their algorithm?
> >
> > W5: In Section 2: Related work, the authors mentioned that "to address the issue [of combining the RCT data and external controls directly will lead to bias], including matching and bias adjustment, power priors, meta-analytic". If these methods are relevant to the approach in this paper, it'd be useful to include a comparison.
> >
> > Q4: The $\hat{\tau}_{aipw} = \tau$ only holds asymptotically, but may differ from $\tau$ for finite sample.

---

> ### Author Response · Authors · 2025-08-07
> **Response to Official Comment by Reviewer uy6o [1/2]**
>
> Dear Reviewer uy6o,
>
> Thank you for your detailed feedback. Below, we provide a point-by-point response to your additional comments.
>
> > **W1:** It would also be helpful if the authors could provide a theoretical characterization of the MSE of their method.
>
> **Response:** Thank you for your helpful suggestions. We agree that providing a theoretical characterization of the MSE would further strengthen this work. **Following your suggestion, we included additional theoretical analysis of the MSE for our method.** Specifically, we first present the variance reduction achieved by leveraging the additional subset of external controls $\mathcal{S}$, as shown in Proposition R1 below.
>
> **Proposition R1.** Under Assumption 1 only,
>  \$\\text{asy.var}(\\hat \\tau\_{\\mathcal{S}}) \\leq \\text{asy.var}(\\hat \\tau\_{\\textup{aipw}})\$. The variance reduction ( \$(N\_{\\mathcal{E}} + N\_{\\mathcal{S}} ) \\{ \\text{asy.var}(\\hat \\tau\_{\\textup{aipw}}) - \\text{asy.var}(\\hat \\tau\_{\\mathcal{S}})\\}\$) is\
>  \\begin{align\*}   \\mathbb E\_{S} \\left \[ \\frac{ \\pi(X) }{q\^2} \\frac{\\text{Var}\_{S}(Y (0)\|X)}{ 1 - e_1(X) } \\frac{ \\mathbb P\_{S}(A=0, R=0\|X) }{ \\mathbb P\_{S}(A=0\|X) } \\right \],\ \\end{align\*}\
>  where \$\\pi(X)\$, \$\\mathbb P\_{S}(A=0, R=0\|X) = \\mathbb P\_{S}(R=0\|X) \\mathbb P\_{S}(A=0\|R=0,X) = 1-\\pi(X)\$, \$\\mathbb P\_{S}(A=0\|X) = 1- e\_{S}(X)\$.
>
>
> According to the result of Theorem 1, this variance reduction presented in Proposition R1 may come at the cost of increased bias given by $\mathbb{E}_{S}[ \frac{R}{q}( \mu_0(X) - m_0(X))]$. Thus, we have the following conclusion.
>
>
> **Proposition R2.** Under Assumption 1, the asymptotic MSE of $\hat \tau_{S}$ is smaller than that of $\hat \tau_{aipw}$, provided that
> \\\[ \\mathbb{E}\_{S} \\left \[ \\frac{ \\pi(X) }{q\^2} \\frac{\\text{Var}\_{S}(Y (0)\|X)}{ 1 - e_1(X) } \\frac{ \\mathbb{P}\_{S}(A=0, R=0\|X) }{ \\mathbb{P}\_{S}(A=0\|X) } \\right \] \> \| \\sqrt{N\_{\\mathcal{E}} + N\_{\\mathcal{S}}} \\cdot \\mathbb{E}\_{S}(\\frac{R}{q}( \\mu_0(X) - m_0(X))) \|\^2. \\\]
>
>
> Proposition 2 provides the condition under which the estimator $\hat \tau_{S}$ has a smaller MSE than $\hat \tau_{\text{aipw}}$. It shows that only when the bias is sufficiently small can the additional use of external controls improve upon the estimator $\hat \tau_{\text{aipw}}$.
>
>
> > **W4:** For the optimal borrowed set, what's the definition of "optimal"? In Section 5, only the loss of a single external control sample is defined. For a set z, should we treat the $\hat \theta_{+z}$
>  as the model parameter after adding the whole set z? If so, how do the authors prove the optimality of their algorithm?
>
> **Response:** Thank you for your comments.
>
> First, we apologize for the lack of clarity on the definition of "optimal". **Below, we provide a detailed description of selecting the "optimal" borrowed set. It consists of two steps:**
>
> * **Step 1:** Based on the ranking of influence scores, we construct a series of nested subsets of external controls, where each subset comprises the top-$K$ most comparable samples (i.e., the top-$K$ samples with the smallest influence scores), see lines 183--185 of the manuscript.
>
> * **Step 2:** Find the optimal $K$ (or the optimal borrowed subset of external controls) that minimizes the MSE based on Theorem 1. Formally, The optimal borrowed subset $\mathcal S$ is defined by
> \\\[\\mathcal S\^\*=\\arg \\min\_{\\mathcal{S}\_k \\in {\\bf S} } \\text{MSE}(\\hat{\\tau}\_{\\mathcal S_k}),\\\]
>       where $\mathcal{\bf S} = \\{ \mathcal S_k: k = 1, ..., N_{\mathcal{O}}\\}$, $\mathcal S_k$ denotes a subset of external controls corresponding to the $k$ smallest influence scores, $N_{\mathcal{O}}$ is the sample size of external controls.
>
> From Figure 2 of the manuscript, we can see that the MSE curve will show a trend of first decreasing and then increasing. Then the lowest point of the curve corresponds to the optimal $K$ with the smallest empirical MSE.
>
> **Second, regarding the issue of set $z$, we would like to kindly remind you that our influence scores are calculated for each  single external control sample, not for a set.** In addition, **this is a key advantage of our method:** Unlike the adaptive lasso-based approach, the influence-based approach does not rely on modeling $\mu_{0, \mathcal{O}}(x)$, and the influence score is defined at the individual level. **This brings a significant merit: the score for each point is unaffected by other points in the external controls, ensuring it is robust to outliers in external controls.** As shown in Figure 1(b), the proposed method effectively selects the top-$K$ points (in red) that are close to the RCT controls, while remaining robust to outliers in the external controls.
>
> Calculating the influence scores at the set level obscures the ability to distinguish which individual samples are truly comparable to the RCT control group and which are not.

---

> ### Author Response · Authors · 2025-08-07
> **Response to Official Comment by Reviewer uy6o [2/2]**
>
> > **W4:** In Section 2: Related work, the authors mentioned that "to address the issue [of combining the RCT data and external controls directly will lead to bias], including matching and bias adjustment, power priors, meta-analytic". If these methods are relevant to the approach in this paper, it'd be useful to include a comparison.
>
> **Response:**  Thank you for your helpful suggestions. We fully agree that including a comparison with more related methods is beneficial. **We summarize the key differences between these methods and ours in the following table.**
>
>
> |                                 | Don't rely on prior |Achieving individual-level sample selection | Model-free to outcome model in external controls| robust to outliers in external controls  |
> |---------------------------------|----------------------------|----------------|---------------------|-----|
> | matching and bias adjustment    |  √                         |×                          | ×                   | ×                   |
> | power priors                    | ×                          |×                          | √                    | ×                   |
> | meta-analytic Predictive Prior  | ×                          |×                          | ×                   | ×                   |
> | lasso-based method Gao et al. (2024) | √                     | √                         | ×                   | ×                   |
> | Ours                            | √                          |√                          | √                   | √                   |
>
>
> **This above table highlights the novelty and unique contributions of our work.** (see more discussion in our response to Office Comment by Reviewer mm3d).
>
>
>
> > **Q4:** The $\hat \tau_{aipw}=\tau$ only holds asymptotically, but may differ from $\tau$ for finite sample.
>
> **Response:**  Thank you for your comments. You are right, the $\hat \tau_{aipw}=\tau$ only holds asymptotically. Nevertheless, this is the semiparametric efficient estimator of $\tau$ based solely on the RCT data, and it represents the most optimal estimator attainable using only the RCT data.
>
>
> We hope that the above clarifications address your concerns. If you have any further questions or concerns, we would be glad to discuss them with you. **In addition, we sincerely invite you to kindly re-evaluate our work.** Thanks again.

---

> > ### Author Response · Authors · 2025-08-09
> > **Request for Further Feedback**
> >
> > Dear Reviewer uy6o,
> >
> > I hope this message finds you well.
> >
> > **I’m writing to kindly follow up and ask whether your concerns have been fully addressed in our latest response.** Your feedback has been invaluable to us, and we’ve made every effort to carefully incorporate your suggestions.
> >
> > If there are any remaining questions or points that need further clarification, we would be more than happy to discuss them in detail. Please don’t hesitate to let us know.
> >
> > Thank you again for your time and thoughtful input.
> >
> > Warm regards,
> >
> > Authors

---

> > ### Comment · Reviewer_uy6o · 2025-08-09
> >
> > I thank the authors for their responses and clarifications.

---

> > > ### Author Response · Authors · 2025-08-09
> > > **Thanks to Reviewer uy6o and Further Feedback Welcome**
> > >
> > > Dear Reviewer uy6o,
> > >
> > > Thanks for your feedback. We sincerely appreciate your time  for evaluating our work.
> > >
> > > If our responses have addressed your concerns, we would be most grateful if you might kindly consider increasing your score to support this work.
> > >
> > > Warm regards,
> > >
> > > Authors

---

### Official Review · Reviewer_mm3d · 2025-07-02

**Clarity:** 3
**Significance:** 2
**Originality:** 2
**Rating:** 4
**Confidence:** 4

**Summary:**

This paper proposes a method to improve treatment effect estimation in RCTs by adaptively borrowing external control samples without relying on the exchangeability assumption. The key innovation is an influence-function-based approach that quantifies each external sample’s comparability to the RCT controls through its impact on the outcome model, ranking samples by influence scores. The method constructs nested candidate borrowing sets and derives a semiparametric efficient estimator combining RCT data with these selected controls. The authors analyze asymptotic bias and variance under violations of exchangeability and propose data-driven selection of the optimal borrowed set by minimizing MSE. Simulations and real-world application (NSW/PSID) demonstrate the method’s superiority over existing approaches, including full and adaptive lasso-based borrowing (Gao et al).

**Questions:**

1. Inference and Confidence Intervals: The paper does not provide clear guidance on how to construct valid confidence intervals for the proposed estimator, especially given the adaptive selection of borrowed samples. Could the authors clarify whether the estimator is regular and root-n consistent? If not, how should practitioners approach uncertainty quantification?

2. Choice of K (Number of Borrowed Samples): While the paper proposes ranking external controls by influence scores, there is no practical guidance on selecting the optimal K. Should practitioners perform a grid search over candidate Ks? Could the authors propose a data-driven or theoretically motivated stopping rule?

3. Beyond Covariate Shift: The paper considers covariate shift (differences in X distributions) in simulations but does not address other forms of distribution shift between RCT and external controls, such as differences in Y|A,X (conditional outcome) or A|X (treatment assignment mechanism). Could the authors discuss the approach’s robustness to these more general types of shift in theory and simulations?

4. Detectability of Bias and Limits of Borrowing: What are the limits on the degree of mismatch between RCT and external controls before the method’s bias detection and correction fail? Could the authors quantify scenarios where differences across data sources become too large for reliable borrowing?

5. Comparison with Existing Methods: In what scenarios might the adaptive lasso-based approach (e.g., Gao et al.) outperform the influence-based method? A discussion of failure modes or trade-offs would help practitioners choose between methods.

6. Related Literature: The introduction could better & more broadly be situated in the literature on generalizability/transportability of causal effects (e.g., work on extending causal inferences to new populations or data integration/fusion, see Dahabreh et al.). Relevant papers also include methods for federated causal inference (e.g., Xiong et al, 2023 SIM; Han et al., JASA 2025), etc.

**Ethical Concerns:**

["NO or VERY MINOR ethics concerns only"]

**Final Justification:**

There have been many papers on combining RCT and observational studies, including in the context of external controls, but this paper proposes a novel idea in quantifying each external sample’s comparability to the RCT controls through its impact on the outcome model by ranking samples via a measure it calls "influence scores." Although there are some limitations to the method, namely in its inference, I find that the authors have engaged with the reviewers and I believe the paper has improved via this process and contributes to the literature via a new idea; hence my increased score.

**Limitations:**

Yes

**Quality:**

3

**Strengths And Weaknesses:**

Strengths:
* The paper addresses a practically important problem of borrowing external controls when exchangeability may not hold.
* The influence-based approach is relatively novel, theoretically grounded, and provides a principled way to assess sample comparability.
* The paper presents rigorous asymptotic analysis of bias and variance.
* Extensive simulations and real-data application demonstrate improvements over existing methods.
* The paper is generally well-organized and clearly motivated, helping readers understand the method’s intuition.

Weaknesses:
* The paper lacks guidance on inference, including how to construct valid confidence intervals for the proposed estimator.
* There is no practical strategy for choosing K, leaving practitioners without a clear selection procedure for the optimal number of borrowed samples.
* The method’s robustness to other distributional shifts (e.g., outcome or treatment assignment mechanisms) beyond covariate shift is not explored.
* The computational burden of influence score calculation, especially for large-scale or high-dimensional data, is not addressed in detail.
* The introduction could better situate the work in the broader literature on generalizability, transportability, and federated causal inference.

---

> ### Author Rebuttal · Authors · 2025-07-29
>
> We thank you for your efforts in evaluating our work. **Below, we hope that our clarification addresses your concerns and sincerely invite you to kindly re-evaluate our work in light of our explanation.**
>
> > **Q1:** Inference and Confidence Intervals: The paper does not provide clear guidance on how to construct valid confidence intervals for the proposed estimator, especially given the adaptive selection of borrowed samples. Could the authors clarify whether the estimator is regular and root-n consistent? If not, how should practitioners approach uncertainty quantification?
>
> > **Response to Q1:** Thanks for your comments.
>
> **Theoretically,** from Proposition 2 and Theorem 1, the proposed estimator is regular and root-$n$ consistent under Assumptions 1 and 2, and may exhibit bias when only Assumption 1 holds. Furthermore, we derive the asymptotic variance in both Proposition 2 and Theorem 1, which can be used to construct confidence intervals for the proposed estimator.
>
> **Empirically,** in our experiments, we presented the standard error (S.E). and the bias of the estimator in Table 3, and we can naturally construct the $95\\%$ confidence interval, by $\text{Est.}\pm 1.96\times\frac{\text{S.E.}}{\sqrt{N_{\mathcal{E}}+N_{\mathcal{S}}}}$, where "Est." represents "Point estimates". We added confidence intervals in the following table.
>
> |  | Top-K=300  |   | Top-K=350  | | Top-K=400     |   |
> |--|--|--|--|--|--|--|
> | Linear | Est.(S.E.) | C.I.| Est.(S.E.)| C.I. | Est.(S.E.) | C.I.  |
> | $\hat{\tau}_{\textup{aipw}}$  | 0.285(0.107) | [0.274,0.296]    | 0.285(0.107)  | [0.274,0.296]    | 0.285(0.107)  | [0.274,0.296]     |
> | $\hat{\tau}_{\textup{full}}$  | 0.19(0.091)   | [0.185, 0.195]   | 0.19(0.091)   | [0.185, 0.195]   | 0.19(0.091)   | [0.185, 0.195]    |
> | $\hat{\tau}_{\textup{lasso}}$ | 0.363(0.101)  | [0.356, 0.371]   | 0.364(0.099)  | [0.357, 0.371]   | 0.339(0.098)  | [0.332, 0.346]    |
> | $\hat{\tau}_{\textup{if}}$    | 0.31(0.08)  | [0.304, 0.316]   | 0.311(0.081)  | [0.304, 0.316]   | 0.293(0.081)  | [0.288, 0.299]    |
> | | | | | | | |
> |  | Top-K=50 | | Top-K=100 | | Top-K=150 |  |
> | Nonlinear | Est.(S.E.) | C.I.    | Est.(S.E.)  | C.I. | Est.(S.E.)    | C.I.              |
> | $\hat{\tau}_{\textup{aipw}}$  | 1.424(0.216)  | [1.404,1.443]    | 1.424(0.216)  | [1.404,1.443]    | 1.424(0.216)  | [1.404,1.443]     |
> | $\hat{\tau}_{\textup{full}}$  | 0.975(0.186)  | [0.964, 0.986]   | 0.975(0.186)  | [0.964, 0.986]   | 0.975(0.0186) | [0.964, 0.986]    |
> | $\hat{\tau}_{\textup{lasso}}$ | 1.392(0.212) | [1.373, 1.412]   | 1.359(0.221)  | [1.340, 1.379]   | 1.305(0.223)  | [1.286, 1.324]    |
> | $\hat{\tau}_{\textup{if}}$    | 1.393(0.211)  | [1.373, 1.413]   | 1.378(0.213)  | [1.359, 1.397]   | 1.376(0.215)  | [1.358, 1.394]    |
> |                               |                          |                  |               |                  |               |                   |
> | | Top-K=30 | | Top-K=35 | | Top-K=45      |                   |
> | NSW \& PSID | Est.(S.E.) | C.I.  | Est.(S.E.)    | C.I.  | Est.(S.E.)    | C.I.              |
> | $\hat{\tau}_{\textup{aipw}}$  | 0.009(0.013)  | [0.007,0.10]     | 0.009(0.013)  | [0.007,0.10]     | 0.009(0.013)  | [0.007,0.10]      |
> | $\hat{\tau}_{\textup{full}}$  | -0.013(0.015) | [-0.015, -0.012] | -0.013(0.015) | [-0.015, -0.012] | -0.013(0.015) | [-0.015, -0.012]  |
> | $\hat{\tau}_{\textup{lasso}}$ | -0.006(0.014)  | [-0.008, -0.005] | -0.008(0.014) | [-0.010, -0.007] | -0.006(0.015) | [-0.008, -0.005]  |
> | $\hat{\tau}_{\textup{if}}$    | 0.006(0.012)  | [0.005, 0.008]   | -0.008(0.013) | [0.006, 0.009]   | 0.010(0.013)  | [0.009, 0.012]    |
>
>
> > **Q2:** Choice of K (Number of Borrowed Samples): While the paper proposes ranking external controls by influence scores, there is no practical guidance on selecting the optimal K. Should practitioners perform a grid search over candidate Ks? Could the authors propose a data-driven or theoretically motivated stopping rule?
>
>  **Response to Q2:** Thanks for your comments. **We would like to kindly remind the reviewer that we have proposed a method for selecting the optimal $K$ by minimizing the MSE of the estimator $\hat \tau_{S}$.  For clarity, we provide a detailed description of selecting the "optimal" borrowed set. It consists of two steps:**
>
> - Step 1:  Based on the ranking of influence scores, we construct a series of nested subsets of external controls, where each subset comprises the top-K most comparable samples (i.e., the top-K samples with the smallest influence scores), see lines 183--185 of the manuscript.
> - Step 2: Find the optimal $K$ that minimizes the MSE based on Theorem 1, see Section 6.2 of the manuscript.
>
> **From Figure 2 of the manuscript, we can see that the MSE curve will show a trend of first decreasing and then increasing. Then the lowest point of the curve corresponds to the optimal K with the smallest empirical MSE.**
>
> > **Q3:** Beyond Covariate Shift: The paper considers covariate shift (differences in X distributions) in simulations but does not address other forms of distribution shift between RCT and external controls, such as differences in Y|A, X (conditional outcome) or A|X (treatment assignment mechanism). Could the authors discuss the approach’s robustness to these more general types of shift in theory and simulations?
>
>  **Response to Q3:**  Thanks for your comments. **We would like to kindly remind the reviewer that we have considered both the differences in $Y|A, X$ and $A|X$. In Table 2 (Simulation setting), the term $ \delta T$ constitutes the differences in Y|A, X (conditional outcome), where $\delta$ represents the level of inconcurrency. We describe this experimental setup in lines 267-268.**
>
> **In addition, we additionally conducted experiments with different $\delta$ to illustrate the robustness of the proposed method.** As shown in the table below, we can see that: (1)the proposed method stably performs well. (2) as $\delta$ increases, the optimal $K$ becomes smaller. This is intuitive because a larger $\delta$ means fewer samples in the external sample that meet comparability requirements.
>
> | | $\delta=0.3$| $\delta=0.5$ | $\delta=0.7$ |
> |--|--|--|--|
> | Linear |  MSE(optimal_K = 300) | MSE(optimal_K = 250)| MSE(optimal_K = 200) |
> | $\hat{\tau}_{\textup{aipw}}$ | 0.0116 | 0.0116 | 0.0116|
> | $\hat{\tau}_{\textup{full}}$ | 0.0807 | 0.2046 | 0.3893 |
> | $\hat{\tau}_{\textup{lasso}}$ |0.0155 | 0.0134 | 0.0132 |
> | $\hat{\tau}_{\textup{if}}$ | **0.0067**| **0.0071** |**0.0067**|
>
> For the A|X (treatment assignment mechanism), the treatment assignment for the trial data is completely at random, and $A = 0$ for all external controls.
>
>
> > **Q4:** Detectability of Bias and Limits of Borrowing: What are the limits on the degree of mismatch between RCT and external controls before the method’s bias detection and correction fail? Could the authors quantify scenarios where differences across data sources become too large for reliable borrowing?
>
>  **Response to Q4:** Thanks for raising this interesting question. **Reliable borrowing of our method depends on the difference between RCT and external control not being too large.
> When
> \\\[      \mathbb{E}[Y(0) | X=x,R=1]  - \mathbb{E}[Y(0) | X=x,R=0]  \text{ is large for all } x,   \\\]
> our method fails.** This is because, in this case, no comparable samples are available between RCT and external control.  All influence scores are large, and introducing the external controls will incur a large bias compared to its contribution in variance reduction. We also would like to clarify that this problem is not unique to our method. All previous methods also fail.
>
> **Moreover, we added additional experiments to illustrate this empirically by setting a large value of $\delta$ (see table below)**. The results show that only a small number of external controls are selected, and the standard AIPW method exhibits the most competitive performance.
> | | $\delta=5$ |
> |-|-|
> |Linear| MSE(optimal_K = NA) |
> |$\hat{\tau}_{\textup{aipw}}$ |**0.0116** (K=10)|
> |$\hat{\tau}_{\textup{full}}$ |44.124(K=10)|
> |$\hat{\tau}_{\textup{lasso}}$|0.036(K=10) |
> |$\hat{\tau}_{\textup{if}}$|0.033(K=10)|
>
>
> > **Q5:**  Comparison with Existing Methods: In what scenarios might the adaptive lasso-based approach (e.g., Gao et al.) outperform the influence-based method? A discussion of failure modes or trade-offs would help practitioners choose between methods.
>
>  **Response to Q5:** Thanks for your suggestions.  **We are not aware of any settings where the lasso-based method is better than ours. If you have further comments on this, we welcome further discussion.** **In addition, we would like to clarify that our method has two key merits compared with Gao et al. (2024): Better comparability and insensitivity to outliers, please refer to Section 4 for more details. These two salient merits stem from the fact that ''our method is model-free for outcome model in the external controls.'**' In contrast, the lasso-based method (Gao et al. (2024)) is model-dependent and relies on the outcome model of the external control data, and the simple difference between the two outcome models cannot fully capture the comparability of the external data, as discussed in Section 4.
>
>
> > **Q6:** Related Literature: The introduction could better & more broadly be situated in the literature on generalizability/transportability of causal effects (e.g., work on extending causal inferences to new populations or data integration/fusion, see Dahabreh et al.). Relevant papers also include methods for federated causal inference (e.g., Xiong et al, 2023 SIM; Han et al., JASA 2025), etc.
>
>  **Response to Q6:** We sincerely thank the reviewer for pointing out these helpful references. We fully agree that it would benefit from a more comprehensive discussion of the broader literature on generalizability/transportability and federated causal inference. We will add several discussions for them in our revised version.

---

> > ### Comment · Reviewer_mm3d · 2025-08-02
> >
> > I thank the authors for their comprehensive responses, but I have two clarifications. I tend toward improving my score contingent on these two updates.
> >
> > First, my Q1 about valid inference was related to your response to my question in Q4. Penalization or regularization methods can be useful and have been previously considered in such settings where the difference is large between the datasets for which combination is desired, see e.g. Cheng and Cai (2021, Arxiv): Adaptive Combination of Randomized and Observational Data or Han et al (2023, NeurIPS): Multiply Robust Federated Estimation of Targeted Average Treatment Effects. These are data-adaptive methods that penalize/down-weight sites that are biased relative to the target site. Thus, I do not fully agree with the authors' statement that "this problem is not unique to our method. All previous methods fail." It would be helpful to see a sketch of a proposed regularization method as an extension of the authors' proposed method when the difference is large, pointing to these existing methods. Second and relatedly, it would be helpful to see the expanded "Related Literature" that the authors propose, to be sure that it is more comprehensive and covers the literature more accurately.

---

> ### Author Response · Authors · 2025-08-04
> **Response to Official Comment by Reviewer mm3d (1/2)**
>
> > **Q1**: Penalization or regularization methods can be useful and have been previously considered in such settings where the difference is large between the datasets for which combination is desired, see e.g. Cheng and Cai (2021) and Han et al. (2023).
> These are data-adaptive methods that penalize/down-weight sites that are biased relative to the target site. Thus, I do not fully agree with the authors ''statement that "this problem is not unique to our method. All previous methods fail."
>
> **Response to Q1:** Thank you for pointing out these relevant references.
>
> **First, we would like to clarify that our setting differs significantly from these references. Below, we summarize the key differences.**
>
> ||Don't rely on exchangeability assumption|Achieving individual-level sample selection|Model-free to outcome model in external controls|
> |-|-|-|-|
> |Han et al. (2023, NIPS)|×|×|×|
> |Han et al. (2025, JASA)|×|×|×|
> |Cheng and Cai. (2021, arXiv)|√|×|×|
> |Gao et al. (2024, Biometrika)|√|√|×|
> |Ours|√ |√|√|
>
> **From this Table, only our method and Gao et al.'s method achieve individual-level selection. This is why we primarily focus on comparing it in the manuscript.**
>
> **Second,  our method addressed the scenario where both covariate shift ($\mathbb P(X\mid R=1) \neq \mathbb P(X \mid R=0)$) and concept shift ($\mathbb P(Y(0)\mid X, R=1) \neq \mathbb P(Y(0)\mid X, R=0) $) are present,  while Han et al. (2023, 2025) only consider the case of ``covariate shift."**  Specifically,
> * The work of Han et al. (2025, JASA) does not consider the case of concept shift, all their theoretical analyses are based on the assumptions of no posterior drift, allowing only for the covariate shift. Please refer to Assumption 1(d) and Section 4 of  Han et al. (2023, NIPS).
> * The work of Han et al. (2023, NIPS) does not consider the case of concept shift, all their theoretical analysis are based on the assumptions of no concept shift, allowing only for the covariate shift. Please refer to Assumption (A6) and Section 5 of  Han et al. (2025, JASA).
>
> Since our main contribution (and motivation) is to avoid the exchangeability assumption (Assumption 2 of the manuscript, $\mathbb P(Y(0)\mid X, R=1) \neq \mathbb P(Y(0)\mid X, R=0)$  or $\mathbb E(Y(0)\mid X, R=1) \neq \mathbb E(Y(0)\mid X, R=0) $), allowing for the presence of concept shift, we do not further compare our method with theirs.
>
> **Third, we provide a detailed comparison between our method and that of Cheng and Cai. (2021, arXiv).**
> Cheng and Cai. (2021, arXiv) considered an interesting setting: if we can obtain  **a consistent estimator**  of the target estimand $\tau$ from RCT data, and obtain a  **biased estimator** from the external data, the authors proposed novel strategies to obtain an enhanced estimator by combining both consistent and biased estimators, by using the regularization technique.
>
> **For clarity, we give a brief description of the method proposed by Cheng and Cai. (2021, arXiv).** We adopt the same notation in our manuscript. Let $X$ be the covariates, $Y$ be the outcome, and $A$ be a binary treatment. In potential outcome framework, we denote $Y(0)$ and $Y(1)$ as the potential outcomes under treatment arms $0$ and $1$, respectively.  Suppose that we have access to RCT data and external data. Let $R$ be the indicator of the data source, where $R=1$ represents the RCT data and $R=0$ represents the external data.  Under a superpopulation model $\mathbb P$, we assume that the data $\{(X_i,A_i,Y_i(0),Y_i(1),R_i),i=1,\ldots,n\}$ are independent and identically distributed for all units in the RCT and external data. The observed datasets are given as
> \\\[ \\begin{cases}\
> \\text{RCT data: }\\{(X_i,A_i,Y_i,R_i = 1), i = 1, ..., n_1\\},
> \\\\\
> \\text{External data: } \\{(X_i, A_i,Y_i,R_i = 0), i = n_1+1, ...,
> n=n_1+n_0\\}\
> \\end{cases}\
> \\\]
> Let $\mathbb E$ be the expectation operator of $\\mathbb P$, then the ATE in the RCT is denoted as
> \$\$\\tau = \\mathbb E\\{Y(1)-Y (0) \\mid R = 1\\}.\$\$
>
> **For estimating $\tau$, the proposed method of Cheng and Cai. (2021, arXiv) consists of two steps:**
>
> Step 1: Construct two estimators of $\tau$, based on each single dataset (RCT or External data).
> Specifically,
> - Consistent estimator $\hat \tau_{\text{aipw}}$:  we use $\hat \tau_{\text{aipw}}$ (our baseline), which is a consistent estimator of $\tau$ based solely on the RCT data.
> - Biased estimator $\hat \tau_{\text{full}}$: If the external data contain both treated and control units, we may also adopt the AIPW estimator based solely on the external data (which may be confounded) as a biased estimator of $\tau$. If the external data contain only the control units (our setting; and Gao et al. (2024)'s setting), we could use $\hat \tau_{\text{full}}$ (our baseline) as the biased estimator of $\tau$.
>
> Since Cheng and Cai. (2021, arXiv) allows for the possibility that $\hat \tau_{\text{full}}$ is biased. This potentially accommodates both covariate shift and concept shift.

---

> ### Author Response · Authors · 2025-08-04
> **Response to Official Comment by Reviewer mm3d (2/2)**
>
> Step 2: Propose a strategy for selecting an optimal weight to combine two estimators in Step 1.
>
> Cheng and Cai. (2021, arXiv) considered a class of weighted estimator
>          \\\[ \\hat \\tau\_{w}:= \\hat \\tau\_{\\text{aipw}} + w \\cdot (\\hat
> \\tau\_{\\text{full}} - \\hat \\tau\_{\\text{aipw}} ), \\\]\
> where \$w\$ is the weight. For determining the weight $w$, Cheng and Cai. (2021, arXiv) recommended selecting the one that minimizing the MSE of $\hat \tau_{w}$, and proposed a regularization method to select the optimal parameter.
>
> **From the procedure of Cheng and Cai. (2021, arXiv)'s method, we can see that it will fail---or the weight$w$ will be zero--- if
>    \\\[ \\mathbb E (Y(0)\\mid X, R=1) \\neq \\mathbb E (Y(0)\\mid X, R=0) \\quad
> \\text{is large for all } X. \\\]
> As noted in the first paragraph of Section 7.1 in Cheng and Cai, (2021 arXiv), ".... subjects with high differences between two datasets..... As a result, the weights are shrunk towards zero
> so that the adaptive estimates are shrunk towards the trial estimates."** **Therefore, we maintain our view that "this problem is not unique to our method."**
>
> > **Q2**: Second and relatedly, it would be helpful to see the expanded "Related Literature" that the authors propose, to be sure that it is more comprehensive and covers the literature more accurately.
>
> **Response to Q2:** Thank you for your helpful suggestions. We will include the following paragraph in this work.
>
> Beyond combining RCT data with external control data, there are many other settings for data combination. For example, one can combine RCT (or experimental) data with external data that contains only covariates Dahabreh et al. (2020 SIM, 2019 Biometrics). In such settings, the goal is typically to generalize the causal effect from the RCT to the external population, rather than to improve causal effect estimation within the RCT itself. Another common setting involves combining RCT data with confounded observational data that suffer from unobserved confounders (Cheng and Cai, 2021 arXiv; Yang et al., 2023 JRSSB). In addition, several studies have combined multiple datasets for causal inference (Han et al., 2025 JASA, 2023 NIPS, Xiong et al. 2023 SIM). In addition, several studies have combined multiple datasets for causal inference (Han et al., 2025 JASA, 2023 NIPS). When combining multiple datasets, it is important not only to consider which identifiability assumptions to adopt in order to improve causal effect estimation, but also to address how to preserve privacy. Unlike these studies that either use the entire external dataset or do not use external data at all, our method performs individual-level selection of external data.
>
> ---
>
> **References**
>
> Cheng, D. and Cai, T. (2021). Adaptive combination of randomized and observational data. arXiv
> preprint 2111.15012.
>
> Dahabreh, I. J., Robertson, S. E., Steingrimsson, J. A., Stuart, E. A., and Hern´an, M. A. (2020).
> Extending inferences from a randomized trial to a new target population. Statistics in Medicine,
> 39:1999–2014.
>
> Dahabreh, I. J., Robertson, S. E., Tchetgen, E. J., Stuart, E. A., and Hern´an, M. A. (2019). Gen-
> eralizing causal inferences from individuals in randomized trials to all trial-eligible individuals.
> Biometrics, 75(2):685–694.
>
> Han, L., Hou, J., Cho, K., Duan, R., and Cai, T. (2025). Federated adaptive causal estimation
> (face) of target treatment effects. Journal of the American Statistical Association.
>
> Han, L., Shen, Z., and Zubizarreta, J. R. (2023). Multiply robust federated estimation of targeted
> average treatment effects. In Conference on Neural Information Processing Systems, number
> 3087, pages 70453–70482.
>
> Yang, S., Gao, C., Zeng, D., and Wang, X. (2023). Elastic integrative analysis of randomised
> trial and real-world data for treatment heterogeneity estimatio. Journal of the Royal Statistical
> Society Series B: Statistical Methodology, 85(3):575–596.
>
> Xiong R, Koenecke A, Powell M, et al. Federated causal inference in heterogeneous observational data[J]. Statistics in Medicine, 2023, 42(24): 4418-4439.

---

> > ### Comment · Reviewer_mm3d · 2025-08-04
> >
> > I thank the authors for their detailed response. However, I have a couple points of clarification. First, the papers by Han et al. do allow for concept shift, see for example, the construction of their regularized estimator, which anchors on the estimator from a target site and estimates a weight vector for the augmented difference relative to the source sites, where the weight vector is data-adaptively selected as the minimizer of a loss function consisting of a MSE term and a bias term (penalty). The exchangeability assumptions made by Han et al are for causal identification of the target estimand from the sources, but the authors allow for concept shift in the actual estimation & inference procedure, and are able to correct for this bias-variance trade-off through the weights, similar to Cheng & Cai. Second, perhaps this is a difference in terminology, but I would not use the term "fail" for shrinking weights to 0 when the bias is large. This is a good property, since otherwise bias would dominate & induce negative transfer. This leads to my main question: Does the authors' method have the ability to data-adaptively exclude the external controls, or must this be determined a priori before the calculation of influence scores? This is an important point of clarification. I am confused by the authors' previous response (in particular, the part in bold), which I repeat below: "This is because, in this case, no comparable samples are available between RCT and external control. All influence scores are large, and **introducing the external controls will incur a large bias** compared to its contribution in variance reduction."

---

> ### Author Response · Authors · 2025-08-04
> **Response to Official Comment by Reviewer mm3d**
>
> We sincerely appreciate the reviewers for their timely feedback. Below, we provide a point-by-point response to your additional comments.
>
> > **Q1**: First, the papers by Han et al. do allow for concept shift, see for example, the construction of their regularized estimator, which anchors on the estimator from a target site and estimates a weight vector for the augmented difference relative to the source sites, where the weight vector is data-adaptively selected as the minimizer of a loss function consisting of a MSE term and a bias term (penalty). The exchangeability assumptions made by Han et al are for causal identification of the target estimand from the sources, but the authors allow for concept shift in the actual estimation \& inference procedure, and are able to correct for this bias-variance trade-off through the weights, similar to Cheng \& Cai.
>
> **Response to Q1**:  Thank you very much for your clear and thoughtful explanation. We fully agree that Han et al.'s method allows for concept shift during estimation and inference. Much appreciated.
>
> > **Q2**: Second, perhaps this is a difference in terminology, but I would not use the term "fail" for shrinking weights to 0 when the bias is large. This is a good property, since otherwise bias would dominate \& induce negative transfer. This leads to my main question: Does the authors' method have the ability to data-adaptively exclude the external controls, or must this be determined a priori before the calculation of influence scores? This is an important point of clarification.
>  I am confused by the authors' previous response (in particular, the part in bold), which I repeat below: "This is because, in this case, no comparable samples are available between RCT and external control. All influence scores are large, and introducing the external controls will incur a large bias compared to its contribution in variance reduction."
>
> **Response to Q2**: Thank you for your clarification and kind explanation. We apologize for the lack of clarity regarding the term 'fail'. In our previous response, we used the term 'fail' to refer to cases where the estimator based solely on the RCT is already optimal, and external data cannot be leveraged to improve it.
>
> **If this is a good property, then our method will have this property.** The reason is as follows: When
> \\\[ \\mathbb E (Y(0)\\mid X, R=1) \\neq \\mathbb E (Y(0)\\mid X, R=0) \\quad
> \\text{is large for all \$X\$ in external controls}, \\\]
> in such a case, our method will yield high influence scores for almost all individuals in external controls. This is because a large bias results in a high influence score: for any $z=(x,y)$ in external controls, a large bias implies that $(y -\hat{\mu}_{0}(x;\theta))$ is large, which, by definition, leads to a high influence score.
>
> After calculating the influence scores for each individual in external controls, we then sort the influence scores, and construct a series of nested subsets of external controls, where each subset comprises the top-$K$ individuals with the smallest influence scores.  Next, we aim to find the optimal $K$ (or the optimal subset of external controls with sample size $K$) that minimizes the MSE of the estimator based on Theorem 1. **If all biases are large, the optimal choice of $K$ will naturally tend toward zero.** This follows the exact same rationale: the bias dominates.
>
> **In addition, and most importantly, the influence score-based method can measure the influence of each individual in external controls and adaptively select the individuals with small influence scores and exclude the individuals with large influence scores.
> This brings a significant merit: It could deal with the following case
>   \\\[ \\mathbb E (Y(0)\\mid X, R=1) \\neq \\mathbb E (Y(0)\\mid X, R=0) \\quad
> \\text{is large for some values of \$X\$ in the external controls and
> small for others.} \\\]**
>
>
> **We hope that the above clarifications address your concerns. If you have any further questions or concerns, we would be glad to discuss them with you.**

---

> > ### Author Response · Authors · 2025-08-07
> > **Official Comment and Further Feedback Welcome**
> >
> > Dear Reviewer mm3d,
> >
> > I hope this message finds you well.
> >
> > **I’m writing to kindly follow up and ask whether your concerns have been fully addressed in our latest response.** Your feedback has been invaluable to us, and we’ve made every effort to carefully incorporate your suggestions.
> >
> > If there are any remaining questions or points that need further clarification, we would be more than happy to discuss them in detail. Please don’t hesitate to let us know.
> >
> > Thank you again for your time and thoughtful input.
> >
> > Warm regards,
> >
> > Authors

---

> > > ### Comment · Reviewer_mm3d · 2025-08-07
> > >
> > > I do not understand the authors' insistence on comments such as, "In contrast, Han et al.'s method cannot achieve this." That method was designed for exactly these type of scenarios in the adaptive estimation step, and it takes this into account for valid inference. It seems that there is still some confusion on the authors' part about causal identification and adaptive estimation. On the other hand, I do not see how their method can accomodate this in the inference. They state that "the optimal choice of K will naturally tend toward zero." But then the inference should take the uncertainty into account for estimating K.

---

> > > > ### Author Response · Authors · 2025-08-07
> > > > **Response to Official Comment by Reviewer mm3d**
> > > >
> > > > Dear Reviewer mm3d,
> > > >
> > > > Thank you for your prompt feedback. We apologize for the lack of clarity, which may have led to some misunderstanding. In response, we removed the sentence "Han et al.'s method cannot achieve this" in our last response.
> > > >
> > > > **Below, we give an intuitive interpretation on our method in an extreme case:
> > > >   \\\[ \\mathbb E (Y(0)\\mid X, R=1) - \\mathbb E (Y(0)\\mid X, R=0) \\quad
> > > > \\text{is infinite for a half of individuals in the external controls and is zero for others.} \\\]**
> > > >
> > > > In this case, the estimator of Cheng and Cai. (2021, arXiv), the most optimal weighted estimator by combining a biased estimator and a consistent estimator), where the biased estimator has an infinite bias. In this case, as we discussed before, the weight will be tend to zero, and the weighted estimator of Cheng and Cai. (2021, arXiv) reduces to the estimator based only the RCT data. That is, **the method of Cheng and Cai (2021, arXiv) cannot leverage the external controls at all in such a exterme case. The Han's method have similar property of Cheng and Cai (2021, arXiv) (as you mentioned before).**
> > > >
> > > > In this extreme case, for our method, **the influence score is defiend for each individual in the external controls. The influence scores will be small for half of the individuals and infinite for the others. Thus, we could use the half of external controls, and thus may potentially have a better performance.**
> > > >
> > > > **We’re sorry if our earlier message was unclear. It was certainly not our intention to cause any discomfort, and we sincerely apologize if it came across that way. We simply hoped to have a clear discussion of the relevant points.**
> > > >
> > > > Thank you again for your time and thoughtful input.
> > > >
> > > > Warm regards,
> > > >
> > > > Authors

---

### Official Review · Reviewer_sU8Q · 2025-07-02

**Clarity:** 3
**Significance:** 3
**Originality:** 3
**Rating:** 4
**Confidence:** 4

**Summary:**

This paper proposes an influence function-based adaptive sample borrowing method to enhance statistical efficiency in randomized controlled trials by incorporating external controls. The paper presents a data-driven selection strategy to optimally manage the bias-variance trade-off under potential exchangeability violation when identifying external control samples for borrowing.

**Questions:**

1. Section 5, line 181 to 186, how to decide the optimal $K$? What if there are ties among the influence scores?
2. In practice, how to determine whether Assumption 3 is violated? Could the authors comment on this?
3. Regarding the top-K selective borrowing approach, is the selected subset consistent? It would be helpful if the authors could provide further explanation or theoretical guarantees regarding the set selection consistency.

**Ethical Concerns:**

["NO or VERY MINOR ethics concerns only"]

**Limitations:**

yes

**Quality:**

3

**Strengths And Weaknesses:**

Strengths:
The paper is clearly written and well-structured. It makes an effort to provide intuitions and examples to explain the motivation and the proposed methodology.

Weakness:
- Novelty: It appears that the proposed method is built upon the semiparametric efficiency results derived in Section 6.1. However, similar results can be found in Li et.al, 2023 (reference 9 cited in the paper). Could the authors provide more discussions on this connection?
- Contributions: The top-k selective borrowing strategy and the relaxation of the exchangeability assumption appear to be the paper's primary contributions, but these points are not sufficiently highlighted. To clarify the selective borrowing component, it might be helpful to explicitly formulating the top-k selection as an objective function, perhaps under Eq. (5)? Regarding the relaxation of the exchangeability assumption, highlighting and elaborating further on lines 235 to 239 could strengthen the presentation of this contribution.
- Theoretical results: It would be helpful to have some theoretical results (e.g., remarks) on the efficiency gain of the proposed methods.

---

> ### Author Rebuttal · Authors · 2025-07-30
>
> Thank you very much for your positive evaluation of our paper. Below, we hope that our clarification addresses your concerns.
>
> > **W1:** Novelty: It appears that the proposed method is built upon the semiparametric efficiency results derived in Section 6.1. However, similar results can be found in Li et.al, 2023 (reference 9 cited in the paper). Could the authors provide more discussion on this connection?
>
> **Response to W1:** Thank you for your insightful comments. We agree that both studies share common ground in the semiparametric efficiency theory framework. However, **we would like to clarify that our primary contributions are not in the estimation itself, but in developing a novel approach for selecting external comparable samples and determining the optimal selection set.**
>
> - In Section 4, we provided a comparison between our influence-based method and the lasso-based approach introduced in [1], as illustrated in Figure 1. The results from our experiments demonstrate that the influence-based method offers greater reliability in selecting external samples (see the Table below), which is the key motivation behind our work.
> - In Sections 5--6, we developed the influence-based method and the optimal borrowed set selection method.
>
> |                                 | model-free for external controls | robust to outliers  |
> |---------------------------------|----------------------------------|---------------------|
> | the adaptive lasso-based method | ×                                | ×                   |
> | the influence-based method      | √                                | √                   |
>
> **In addition, in lines 61--68 (the end of Introduction) of the manuscript, we did not claim any contribution to semiparametric efficiency. We only claimed the following main contributions:**
> - We reveal the limitations of existing approaches for estimating treatment effects by combining RCT data with external controls.
> - We propose an influence-based sample borrowing approach, which can effectively quantify the comparability of each sample in the external controls.
> -  We develop a data-driven approach to select the optimal subset of external control samples based on the MSE of the proposed estimator.
> - We conduct extensive experiments on both simulated and real-world datasets, demonstrating that the proposed approach outperforms the existing baseline approaches.
>
> We hope that the above clarification could address your concerns.
>
>
> > **W2:** Contributions: The top-k selective borrowing strategy and the relaxation of the exchangeability assumption appear to be the paper's primary contributions, but these points are not sufficiently highlighted. To clarify the selective borrowing component, it might be helpful to explicitly formulate the top-k selection as an objective function, perhaps under Eq. (5)? Regarding the relaxation of the exchangeability assumption, highlighting and elaborating further on lines 235 to 239 could strengthen the presentation of this contribution.
>
> **Response to W2:** Thank you for your comments. We appreciate your suggestions to emphasize the selective borrowing strategy and the relaxation of the exchangeability assumption. To this end, we can explicitly formulate the top-k selective borrowing strategy as an objective function as follows:
> \$\$S\^\*=\\arg \\min\_{S_k \\in \\mathcal{C} } MSE (\\hat{\\tau}\_{S_k}),\$\$
> where \$\\mathcal{C} = \\{ C_k: k = 1, ..., N\_{O}\\}\$, $C_k$ denotes a subset of external controls corresponding to the $k$ smallest influence function values, $N_{O}$ is the sample size of external controls, and $\mathcal{S}^*$ is the optimal candidate. Regarding the relaxation of the exchangeability assumption, we will revise the manuscript to further elaborate on it. Thanks again.
>
>
> > **W3:**  Theoretical results: It would be helpful to have some theoretical results (e.g., remarks) on the efficiency gain of the proposed methods.
>
> **Response to W3:** Thanks for your helpful suggestions. Under Assumption 1 and Assumption 2 (the exchangeability assumption), Li et al. (2023) [ref. 9 in the manuscript] showed that the estimator $\hat \tau_{S}$, where $S$  is the entire external control dataset,  has a smaller asymptotic variance than $\hat \tau_{\text{aipw}}$.
>
> Likewise, **under Assumptions 1 and 3, by a similar proof, we can show that $\hat \tau_{S}$ (for $S$ satisfies Assumption 3),  has a smaller asymptotic variance than $\hat \tau_{\text{aipw}}$. And the efficiency gain is given as**  \\begin{align\*}\ \\mathbb{E}\_{\\mathcal{S}} \\left \[ \\frac{ \\pi(X) }{q\^2} \\frac{\\text{Var}(Y (0)\|X)}{ 1 - e_1(X) } \\frac{ \\mathbb{P}\_{\\mathcal{S}}(A=0, R=0\|X) }{ \\mathbb{P}\_{\\mathcal{S}}(A=0\|X) } \\right \],\ \\end{align\*} where \$\\mathbb{P}\_{\\mathcal{S}}(A=0, R=0\|X) = \\mathbb{P}\_{\\mathcal{S}}(R=0\|X) \\mathbb{P}\_{\\mathcal{S}}(A=0\|R=0,X) = 1-\pi(X)\$.
>
> **However, when only Assumption 1 holds--allowing for arbitrary differences between the RCT and external controls--$\hat \tau_{S}$ may be biased, and any efficiency gain is not guaranteed. This is exactly the setting we explored.**
>
>
> > **Q1:** Section 5, lines 181 to 186, how to decide the optimal $K$? What if there are ties among the influence scores?
>
> **Response to Q1:** Thanks for your comments, and we apologize for the lack of clarity. **Below, we provide a detailed description of selecting the "optimal" borrowed set. It consists of two steps:**
> - **Step 1:**  Based on the ranking of influence scores, we construct a series of nested subsets of external controls, where each subset comprises the top-K most comparable samples (i.e., the top-K samples with the smallest influence scores), see lines 183--185 of the manuscript.
> - **Step 2:** Find the optimal $K$ that minimizes the MSE based on Theorem 1, see Section 6.2 of the manuscript.
> From Figure 2 of the manuscript, we can see that the MSE curve will show a trend of first decreasing and then increasing. Then the lowest point of the curve corresponds to the optimal K with the smallest empirical MSE.
>
> **In addition, our method also works if there are ties among the influence scores.** In this case, there will be multiple optimal $K$, but they all have the same MSE. This does not affect the associated numerical results.
>
>
> > **Q2:** In practice, how to determine whether Assumption 3 is violated? Could the authors comment on this?
>
> **Response to Q2:** Thank you for raising this interesting question. **We would like to clarify that the proposed method does not rely on Assumption 3. Specifically,**
> - The proposed method is mainly based on Theorem 1, where we derive the estimator’s bias and variance without Assumption 3 (i.e., under Assumption 1 only);
> - From Theorem 1, we could find the appropriate subset $\mathcal{S}$ by minimizing the MSE, and then construct the estimator $\hat\tau_{\mathcal{S}}$ based on it.
>
> **In addition, Assumption 3 is plausible when $\mathcal{S}$ is appropriately selected by the influence function.** As shown in Figure 1(b), the controls in RCT (blue points) and in $\mathcal{S}$ (red points) are close, and follow a similar distribution, and thus Assumption 3 is plausible.
>
> **Furthermore, since we select the $K$ samples with small influence scores, we determine the $K$ by minimizing MSE. This design is subtle and guarantees that the selected $K$ samples in external controls and the RCT's controls are close, rendering Assumption 3 close to holding or only slightly violated.** We could intuitively understand it from the following two aspects:
> - By the definition of influence score, when a sample in external controls is far from the controls in RCT, the influence score tends to be large.
> -  As $K$ increases, if several incomparable samples (the samples are not close to the controls in RCT) appear, then these incomparable samples will yield a bias in the final estimator $\hat\tau_{\mathcal{S}}$, and increase the MSE.
>
>
> > **Q3:**  Regarding the top-K selective borrowing approach, is the selected subset consistent?
>
> **Response to Q3:** Thank you for your insightful comments. We will further explore the issue of selection consistency. Additionally, we would like to clarify that our empirical results show that the proposed method significantly outperforms existing approaches. We agree that the theoretical aspect of selection consistency merits further investigation, and we will include a discussion on this point in the revised manuscript.

---

> > ### Comment · Reviewer_sU8Q · 2025-08-05
> >
> > Thank you for addressing my questions and comments.

---

> ### Author Response · Authors · 2025-08-05
> **Response to Comment by Reviewer sU8Q and Welcome Further Feedback**
>
> We sincerely thank you for your timely feedback. We are glad to know that your concerns have been addressed. If you have any further comments, please feel free to contact us. We welcome any further discussion.
>
> **In this work, we tackle a highly challenging task: developing an individual-level adaptive data-borowwing framework that satisfies several notable (salient) advantages, as summarized in the table below.**
>
> |                                 | Don't rely on exchangeability assumption | Achieving sample-level sample selection  | Model-free to outcome model in external controls|Robust to outliers in external controls |
> |---------------------------------|----------------------------------|---------------------|------------------| ----------------|
> | Li et al. (2023, Biometrics) | ×   | × | × | × |
> | Gao et al. (2024, Biometrika) | √   | √ | × | × |
> | Ours | √   | √ | √ | √ |
>
>
> **This table highlights the novelty and unique contributions of our work.** We believe that our method offers meaningful contributions.
>
> In light of the challenging nature of the problem, the unique contributions, the detailed responses, **we sincerely invite you to kindly re-evaluate our work, and would be deeply grateful if you could generously consider raising our score.** Thanks again.

---

### Official Review · Reviewer_ctUy · 2025-07-04

**Clarity:** 4
**Significance:** 3
**Originality:** 3
**Rating:** 4
**Confidence:** 4

**Summary:**

This paper introduces an influence-function-based framework for adaptively borrowing external control samples to improve treatment effect estimation in RCTs, particularly when sample sizes are small. The method assigns influence scores to external samples based on their perturbation to the RCT control model, allowing selection of the most comparable data without assuming full exchangeability. A semiparametric efficient estimator is then developed, and a data-driven approach selects the optimal subset of external controls by minimizing mean squared error. Through both simulation and real-world experiments, their proposed method demonstrates superior accuracy and robustness compared to existing approaches including adaptive lasso-based borrowing.

**Questions:**

it would be helpful to clarify 1) the influencial validity after adaptive selection; 2) how are the influence score estimation implemented in practice especially in high dimensional settings; 3) provide justification for the approximation of bias as the difference from the AIPW estimator, which tends to have high variability in small samples and hence unreliable.

**Ethical Concerns:**

["NO or VERY MINOR ethics concerns only"]

**Final Justification:**

The authors answered most of my questions but I believe that my original score reflects my overall assessment.

**Limitations:**

There is a notable limitation in their inference - although the author acknowledges a bias-variance-trade-off due to the selection of the comparable subset, ignoring this selection or simply minimizing it may not adequately address the post-selection-inference issue. There is no formal accounting for the variability introduced by the selection part. The theoretical results are derived as if the subset were fixed. The paper sidesteps this with asymptotics and MSE estimation, a rigorous treatment would require additional methods or caveats.

the simulation settings are overly simplistic - in most RCTs, we observe complex covariates including counts, skewed distributions. It would also be important to consider heterogeneous treatment effects. The authors should compare to other methods such as AIPW, causal transfer or domain adaptation approaches. Authors should consider more modern real datasets such as UK Biobank or MIMIC.

**Paper Formatting Concerns:**

no concerns.

**Quality:**

3

**Strengths And Weaknesses:**

The overall quality of the work is high. The authors provide both theoretical analysis and empirical results, establishing consistency, asymptotic normality, and semiparametric efficiency of their proposed estimator. They extend their results to accommodate violations of the exchangeability assumption, and use a bias-variance trade-off to drive a data-adaptive selection of borrowed samples. The experimental results are thorough, and the method demonstrates robustness across different covariate shifts and RCT sample sizes. One limitation is the computational burden associated with calculating influence scores, particularly for high-dimensional models requiring second-order derivatives. While this is acknowledged, a deeper exploration of scalability and approximate alternatives would strengthen the practical applicability of the method.

The clarity of the manuscript is generally strong. The structure is logical, with clear motivation, definitions, and progression through theoretical development and empirical evaluation. Mathematical notation is precise, though occasionally dense, and might benefit from simplified exposition in a few sections. Visuals such as borrowing diagrams and performance plots are well-designed and help convey the main results effectively. Implementation details and reproducibility criteria are addressed in full, with code and data availability confirmed.

The significance of the work is reasonably high. The method targets a well-known limitation in modern trial design and real-world evidence generation—the inability to use external data when the exchangeability assumption is questionable. By moving beyond global assumptions to local, sample-level borrowing decisions, the authors propose a strategy with clear practical implications for trials in rare diseases, oncology, and digital experimentation, where small sample sizes are common. However, the application remains focused on a single classical dataset (NSW/PSID). Demonstration of the method in more contemporary, high-dimensional settings such as electronic health records or biobanks would further enhance its relevance.

The originality of the paper is also fine. The use of influence functions to guide data borrowing in causal inference settings is a novel contribution that differentiates this work from prior lasso-based and bias-penalized approaches. The estimator unifies ideas from semiparametric efficiency theory, causal inference, and robust learning, and contributes new tools to an important methodological frontier. The method has the potential to inspire further work in areas such as transfer learning for treatment effect estimation or adaptive trial design.

In summary, this paper makes some contribution to the literature on treatment effect estimation and external data integration. It presents a theoretically grounded, empirically validated, and practically relevant methodology.

---

> ### Author Rebuttal · Authors · 2025-07-30
>
> Thank you very much for your positive evaluation of our paper. Below, we hope that our clarification addresses your concerns.
>
> > **W1:** One limitation is the computational burden associated with calculating influence scores, particularly for high-dimensional models requiring second-order derivatives. While this is acknowledged, a deeper exploration of scalability and approximate alternatives would strengthen the practical applicability of the method.
>
> **Response to W1:** We sincerely appreciate your insightful comments regarding the computational challenges associated with influence score calculation in high-dimensional settings. We acknowledge this limitation and would like to elaborate on both the challenges and potential solutions:
>
> * **Computational Challenges in High-Dimensional Models:**
> The calculation of influence functions indeed faces significant computational and memory burdens when dealing with high-dimensional models, particularly due to the need for second-order derivative computations. As noted by the reviewer, these computational costs scale quadratically with parameter dimensionality, becoming especially prohibitive for large-scale models such as deep neural networks.
>
> * **Scalability Solutions and Approximation Methods:**
> To address these challenges, the research community has developed several efficient approximation approaches, including:
>   - Iterative methods: Conjugate Gradients (CG) for Hessian-vector products [1]
>   - Stochastic estimation techniques [2,3]
>   - Linear-time Stochastic Second-Order Algorithm (LISSA) [4]
>   - Kronecker-Factored Approximate Curvature (K-FAC) for efficient curvature approximation[5]
>   - Diagonal Hessian approximation [6]
>
> These methods can significantly reduce the computational overhead while maintaining reasonable accuracy in influence function estimation. In future work, we plan to implement and evaluate these approximation techniques to improve the scalability of our method for large-scale models.
>
>
> > **W2:** However, the application remains focused on a single classical dataset (NSW/PSID). Demonstration of the method in more contemporary, high-dimensional settings such as electronic health records or biobanks would further enhance its relevance.
>
> **Response to W2:** Thanks for your helpful suggestions. The influence function method has demonstrated its applicability to high-dimensional datasets empirically, such as images. While our work focuses on different objectives, a notable example is Pang et al. (2017)[3], who investigated the role of influence functions in interpreting black-box models on the MNIST dataset and other image datasets. We will explore high-dimensional settings in future work. Thanks again.
>
> > **W3:**  There is a notable limitation in their inference - although the author acknowledges a bias-variance-trade-off due to the selection of the comparable subset, ignoring this selection or simply minimizing it may not adequately address the post-selection-inference issue. There is no formal accounting for the variability introduced by the selection part. The theoretical results are derived as if the subset were fixed. The paper sidesteps this with asymptotics and MSE estimation;  a rigorous treatment would require additional methods or caveats.
>
> **Response to W3:** Thank you for your insightful comments and helpful suggestions. We acknowledge that the post-selection process could theoretically influence the results. Accordingly, we will add a discussion on post-selection as a limitation of our work.
>
> > **W4:** The authors should compare to other methods such as AIPW, causal transfer or domain adaptation approaches. Authors should consider more modern real datasets, such as the UK Biobank or MIMIC.
>
> **Response to W4:** Thank you for your helpful suggestions. We would like to kindly remind the reviewer that we have included AIPW as one of the baselines in our experiments. AIPW (based only on the RCT data) is the standard baseline (Lines 286-287), which serves as an asymptotically unbiased benchmark for $\tau$ (line 107). In addition, we also included the semiparametric efficient causal transfer method (i.e., $\hat \tau_{\text{full}}$) as the baseline, which relies on the exchangeability assumption (Assumption 2).
>
> **Following your suggestions, we conducted additional experiments using the MIMIC and eICU datasets**, as outlined below.
> - **Data description:**
> The MIMIC dataset [7] consists of $N_\mathcal{E}=2124$ samples with $N_t=902$ in the treated group and $N_c=1222$ in the control group. This dataset contains a treatment indicator (i.e., vasopressor treatment indicator, A: 1 for treated, 0 otherwise), covariates ( age, gender, weight, temperature, glucose, creatinine, bun, wbc, pco2, ph, chloride, potassium, sodium, spo2, heartrate1, and so on). We take the log of the cumulative balance as the interesting outcome $Y$.
> The external dataset, eICU dataset [8], is used for external comparison with the MIMIC dataset, which includes the same structure as the MIMIC.  We only use the control group data (i.e., $A=0$) of a as the external controls ($N_{\mathcal{O}}=2595$).
> - **Results:** Due to space limitations, we present the results near the optimal point in the table below (e.g., K=200, 300). We can see that our proposed method consistently outperforms other borrowing methods.
>
> | MIMIC \& eICU ($10^{-3}$)     | Top-K=200 | Top-K=300  |
> |---------|-----------|------------|
> | $\hat{\tau}_{\textup{aipw}}$  | 0.069     | 0.069      |
> | $\hat{\tau}_{\textup{full}}$  | 1.221     | 1.221      |
> | $\hat{\tau}_{\textup{lasso}}$ | 0.599     | 0.295      |
> | $\hat{\tau}_{\textup{if}}$    | **0.062** | **0.063**  |
>
>
>
> > **Q:** It would be helpful to clarify
>
> >(1) the influential validity after adaptive selection;
>
> >(2) how is the influence score estimation implemented in practice, especially in high-dimensional settings;
>
> >(3) provide justification for the approximation of bias as the difference from the AIPW estimator, which tends to have high variability in small samples and is hence unreliable.
>
> **Response to Q(1):** Please see our response to W3;
>
> **Response to Q(2):** Please see our response to W1;
>
> **Response to Q(3):** Thanks for your insightful comments. Below, we show that $\hat \tau_{\mathcal{S}} - \hat \tau_{\text{aipw}}$ is a consistent estimator of the bias.
>
> Under the conditions of Theorem 1,\
> \\begin{align\*}\
> \\hat \\tau\_{\\mathcal{S}} ={}& \\frac{1}{ N\_{\\mathcal{E}} +
> N\_{\\mathcal{S}} } \\sum\_{i\\in \\mathcal{E}\\cup\\mathcal{S}}
> \\frac{\\pi(X_i)}{q} \\left \[ \\frac{ R_iA_i(Y_i - m_1(X_i))}{
> e\_{\\mathcal{S}}(X_i) } - \\frac{(1-A_i)(Y_i - m_0(X_i)) }{1 -
> e\_{\\mathcal{S}}(X_i)} \\right \] \\\\\
> {}&+ \\frac{1}{ N\_{\\mathcal{E}} + N\_{\\mathcal{S}} } \\sum\_{i\\in
> \\mathcal{E}\\cup\\mathcal{S}} \\frac{R_i}{q}\\{ m_1(X_i) -
> m_0(X_i)\\} + o\_{P}(n\^{-1/2}),\
> \\end{align\*}\
> where \$e\_{\\mathcal{S}}(X) = e_1(X) \\pi(X)\$. In addition, by a similar proof, let \$q =
> N\_{\\mathcal{E}}/(N\_{\\mathcal{E}}+N\_{\\mathcal{S}})\$, we have\
> \\begin{align\*}\
> \\hat \\tau\_{\\textup{aipw}} ={}& \\frac{1}{N\_{\\mathcal{E}}}
> \\sum\_{i\\in \\mathcal{E}} \\left \[ \\frac{ A_i\\{Y_i -
> \\mu_1(X_i)\\}}{e_1(X_i)} - \\frac{(1-A_i)\\{Y_i - \\mu_0(X_i)\\}}{1 -
> e_1(X_i)} + \\{ \\mu_1(X_i) - \\mu_0(X_i) \\} \\right \] +
> o\_{P}(n\^{-1/2}) \\\\\
> ={}& \\frac{1}{ N\_{\\mathcal{E}} + N\_{\\mathcal{S}} } \\sum\_{i\\in
> \\mathcal{E}\\cup\\mathcal{S}} \\frac{R_i}{q} \\left \[ \\frac{
> A_i(Y_i - \\mu_1(X_i))}{ e\_{1}(X_i) } - \\frac{(1-A_i)(Y_i -
> \\mu_0(X_i)) }{1 - e\_{1}(X_i)} \\right \] \\\\\
> {}&+ \\frac{1}{ N\_{\\mathcal{E}} + N\_{\\mathcal{S}} } \\sum\_{i\\in
> \\mathcal{E}\\cup\\mathcal{S}} \\frac{R_i}{q}\\{ \\mu_1(X_i) -
> \\mu_0(X_i)\\} + o\_{P}(n\^{-1/2}).\
> \\end{align\*}\
> Since \$m_1(X) = \\mu_1(X)\$,\
> \\begin{align\*}\
> \\hat\\tau\_{\\mathcal{S}} - \\hat \\tau\_{\\text{aipw}}\
> ={}& \\frac{1}{ N\_{\\mathcal{E}} + N\_{\\mathcal{S}} } \\sum\_{i\\in
> \\mathcal{E}\\cup\\mathcal{S}} \\left \[ \\frac{R_i}{q}
> \\frac{(1-A_i)(Y_i - \\mu_0(X_i)) }{1 - e\_{1}(X_i)} -
> \\frac{\\pi(X_i)}{q} \\frac{(1-A_i)(Y_i - m_0(X_i)) }{1 -
> e\_{\\mathcal{S}}(X_i)} \\right \] \\\\\
> {}&+ \\frac{1}{ N\_{\\mathcal{E}} + N\_{\\mathcal{S}} } \\sum\_{i\\in
> \\mathcal{E}\\cup\\mathcal{S}} \\frac{R_i}{q}\\{ \\mu_0(X_i) -
> m_0(X_i)\\} + o\_{P}(n\^{-1/2}) \\\\\
> ={}& A\_{1n} + A\_{2n} + o\_{P}(n\^{-1/2}),\
> \\end{align\*}\
> We can see that the first term \$A\_{1n}\$ converges to zero at a rate
> of order \$1/\\sqrt{N\_{\\mathcal{E}} + N\_{\\mathcal{S}}}\$, and the second
> term \$A\_{1n}\$ converges to the bias \$\\mathbb{E}[ R (\\mu_0(X) -
> m_0(X))/q]\$. Thus, \$ \\hat\\tau\_{\\mathcal{S}} - \\hat
> \\tau\_{\\text{aipw}}\$ is a consistent estimator of the bias.
>
> ---
>  **References**
>
> [1] Martens J. Deep learning via hessian-free optimization[C]//Icml. 2010, 27: 735-742.
>
> [2] Agarwal N, Bullins B, Hazan E. Second-order stochastic optimization for machine learning in linear time[J]. Journal of Machine Learning Research, 2017, 18(116): 1-40.
>
> [3] Pang Wei Koh and Percy Liang. Understanding black-box predictions via influence functions. In International conference on machine learning, pages 1885–1894. PMLR, 2017.
>
> [4] Agarwal N, Allen-Zhu Z, Bullins B, et al. Finding approximate local minima faster than gradient descent[C]//Proceedings of the 49th Annual ACM SIGACT Symposium on Theory of Computing. 2017: 1195-1199.
>
> [5] Grosse R, Martens J. A kronecker-factored approximate fisher matrix for convolution layers[C]//International Conference on Machine Learning. PMLR, 2016: 573-582.
>
> [6] Guo H, Rajani N F, Hase P, et al. Fastif: Scalable influence functions for efficient model interpretation and debugging[J]. arXiv preprint arXiv:2012.15781, 2020.
>
> [7] Singer et al (2016). The third international consensus definitions for sepsis and septic shock (sepsis-3). JAMA, 315(8):801–810
>
> [8] Chu et al (2023). Targeted optimal treatment regime learning using summary statistics. Biometrika.

---

> ### Author Response · Authors · 2025-08-05
> **Response to Comment by Reviewer ctUy and Further Feedback Welcome**
>
> We sincerely thank you for your timely feedback.
>
> **In this work, we tackle a highly challenging task: developing an individual-level adaptive data-borowwing framework that satisfies several notable (salient) advantages, as summarized in the table below.**
>
> |                                 | Don't rely on exchangeability assumption | Achieving sample-level sample selection  | Model-free to outcome model in external controls|Robust to outliers in external controls |
> |---------------------------------|----------------------------------|---------------------|------------------| ----------------|
> | Li et al. (2023, Biometrics) | ×   | × | × | × |
> | Gao et al. (2024, Biometrika) | √   | √ | × | × |
> | Ours | √   | √ | √ | √ |
>
>
> **This table highlights the novelty and unique contributions of our work.** We believe that our method offers meaningful contributions.
>
> In light of the challenging nature of the problem, the unique contributions of our work, the detailed responses provided, and the additional experiments conducted, **we sincerely invite you to kindly re-evaluate our work, and would be deeply grateful if you could generously consider raising our score.** Thanks again.

---

### Official Review · Reviewer_NK6K · 2025-07-07

**Clarity:** 3
**Significance:** 2
**Originality:** 2
**Rating:** 3
**Confidence:** 4

**Summary:**

This paper proposes an algorithm that borrows information from external controls using an influence-based criterion to improve the estimation of treatment effects. Some theoretical results are developed for the proposed algorithm, and its numerical performance is evaluated through a synthetic example and a real-data application. Overall, the paper is well written and the presentation is clear.

**Questions:**

1. Assumption 3 appears to be very strong. It essentially assumes that the selection procedure can identify unbiased external controls with probability tending to one. However, the paper does not demonstrate that the sample-borrowing procedure described in Section 5 guarantees this assumption in practice. Given this, the resulting theoretical results in Section 6.1 seem to reduce to those for a standard AIPW estimator with an extended control set, which limits their novelty and practical value.

2. Although Assumption 3 is relaxed in Theorem 1 (Section 6.2), the theory still treats the candidate set $\mathcal{S}$ as deterministic. In reality, the selection process for $\mathcal{S}$ is stochastic. Ignoring this stochasticity may lead to an inaccurate or overly optimistic theoretical characterization of the estimator’s properties.

3. How was the propensity score estimated in the numerical experiments? Introducing external controls could potentially complicate the estimation of the propensity score, which may in turn increase the variability of the proposed borrowing algorithm. Some discussion or diagnostic results would help clarify this point.

4. In Figures 2 and 3, why not compare the proposed algorithm to the standard AIPW estimator that does not use external controls? This would provide a clearer benchmark to evaluate the actual benefit of borrowing external information.

5. In Table 3, why is there no comparison of mean squared error (MSE) across different estimators? In several cases, it appears that the proposed estimator performs no better — or even worse — than the regular AIPW estimator.

6. In the experiments, there are 300 treatment units and 100 controls, so it is reasonable to expect that borrowing additional external controls may be beneficial. However, how would the method perform if the number of controls already exceeds the number of treatment units — for example, 100 treatment units versus 300 controls? Would you still expect any improvement in the estimation of the treatment effect? It would be useful to demonstrate, either theoretically or numerically, how the method performs in this setting. Understanding such limitations is important for practical applications.

**Ethical Concerns:**

["NO or VERY MINOR ethics concerns only"]

**Final Justification:**

I have read all reviews/response and would like to keep my score as is.

**Limitations:**

Yes.

**Quality:**

3

**Strengths And Weaknesses:**

Strength: the paper is well-written and the presentation is clear.
Weakness: the theory part seems a bit weak.

---

> ### Author Rebuttal · Authors · 2025-07-29
>
> We thank you for your efforts in evaluating our work. Below, we hope to address your concerns and questions.
>
> > **Q1:** Assumption 3 appears to be very strong. However, the paper does not demonstrate that the sample-borrowing procedure described in Section 5 guarantees this assumption in practice. Given this, the resulting theoretical results in Section 6.1 seem to reduce to those for a standard AIPW estimator, which limits their novelty.
>
> **Response to Q1:**  Thank you for your comments. **We would like to kindly remind the reviewer that there may have been a misunderstanding regarding our method.**
>
> **First, Assumption 3 is plausible when $\mathcal{S}$ is appropriately selected by the influence function.** As shown in Figure 1(b), the controls in RCT (blue points) and in $\mathcal{S}$ (red points) are close, and follow a similar distribution, and thus Assumption 3 is plausible.
>
> **Second and most importantly, the proposed method does not rely on Assumption 3.** Specifically,
> - The proposed method is mainly based on Theorem 1, where we derive the estimator’s bias and variance without Assumption 3 (i.e., under Assumption 1 only);
> - From Theorem 1, we could find the appropriate/optimal subset $\mathcal{S}$ by minimizing the MSE of the estimator $\hat\tau_{\mathcal{S}}$.
>
> **Third, since we select the $K$ samples with small influence scores to form the candidate subset, and we determine the $K$ by minimizing MSE. This design is subtle and guarantees that the selected $K$ samples in external controls and the RCT's controls are close, rendering Assumption 3 holds or close to holds or only slightly violated.** We could intuitively understand it from the following two aspects:
> - By the definition of influence score, when a sample in external controls is far from the controls in RCT, the influence score tends to be large.
> -  As $K$ increases, if several incomparable samples (the samples are not close to the controls in RCT) appear, then these incomparable samples will yield a bias in the final estimator $\hat\tau_{\mathcal{S}}$, and increase the MSE.
> From the above discussion, our proposed method ensures that Assumption 3  is close to holding. This is why we begin our discussion with Assumption 3 in Section 6.1.
>
> **Fourth, for the novelty of Section 6.1 of this article, we would like to clarify that Section 6.1 is not the main contribution of this article.** Section 6.1 is just the beginning of our method. In lines 61--68 (the end of Introduction) of this paper, we only claimed the following main contributions:
> - We reveal the limitations of existing approaches for estimating treatment effects by combining RCT data with external controls.
> - We propose an influence-based sample borrowing approach, which can effectively quantify the comparability of each sample in the external controls.
> -  We develop a data-driven approach to select the optimal subset of external control samples based on the MSE of the proposed estimator.
> - We conduct extensive experiments on both simulated and real-world datasets, demonstrating that the proposed approach outperforms the existing baseline approaches.
>
> **We hope that the above clarification addresses your concerns and sincerely invite you to kindly re-evaluate our work in light of our explanation.**
>
>
> > **Q2:** Although Assumption 3 is relaxed in Theorem 1 (Section 6.2), the theory still treats the candidate set  $S$ as deterministic. In reality, the selection process for $S$  is stochastic.
>
> **Response to Q2:** Thank you for your insightful comments. We acknowledge that the selection stochasticity could theoretically influence the results. Nevertheless, we would like to clarify that our empirical findings consistently show that the proposed method significantly outperforms existing approaches across various settings. We agree that this theoretical aspect warrants further investigation, particularly regarding the uncertainty introduced by the selection process. Accordingly, we added a discussion on selection stochasticity as a limitation of our work.
>
> > **Q3:** How was the propensity score estimated in the numerical experiments? Introducing external controls could potentially complicate the estimation of the propensity score.
>
> **Response to Q3:** Thanks for your comments, and we apologize for the lack of details. By the definition of $e_{\mathcal{S}}(X) = \mathbb{P}_{\mathcal{S}}(A=1|X)$, in our experiments, we use logistic regression to estimate it based $\mathcal{E}\cup \mathcal{S}$. Please see our codes in the Supplementary Material.
>
>  > **Q4:** In Figures 2 and 3, why not compare the proposed algorithm to the standard AIPW estimator that does not use external controls?
>
> **Response to Q4:** Thanks for your comments. In Table 3, we compared the proposed algorithm to the standard AIPW estimator in terms of |bias|  and S.E. (standard error). From Table 3, we can see that the standard error of the standard AIPW estimator is larger than all the methods using external controls.
> **Following your suggestions, we calculate the MSE of the standard AIPW estimator (i.e., $\text{S.E.}^2$ due to zero bias) and compare it with our method, the associated results are given below.**
>
> |Linear|Top-K=300|Top-K=350|Top-K=400|
> |--|--|--|--|
> |$\hat{\tau}_{aipw}$|0.0115|0.0115|0.0115|
> |$\hat{\tau}_{if}$|**0.0071**|**0.0072**|**0.0066**|
> |Nonlinear|Top-K=50|Top-K=100|Top-K=150|
> |$\hat{\tau}_{aipw}$|0.0498|0.0498| 0.0498|
> |$\hat{\tau}_{if}$| **0.0458**|**0.0477**| **0.0487**|
> |NSW \& PSID|Top-K=30|Top-K=35|Top-K=45|
> |$\hat{\tau}_{aipw}$| 0.000181|0.000181|0.000181|
> |$\hat{\tau}_{if}$|**0.000174**|**0.000171**| **0.000174**|
>
> From this table, we could see that the MSE of the standard AIPW estimator is significantly larger than that of the proposed methods.
> In Figures 2 and 3, we don't compare the  standard AIPW estimator in terms of MSE for the following two reasons:
> - Our goal is to compare various competing methods for sample selection, while the AIPW estimator does not conduct sample selection.
> - Visual clarity: For optimal graphical clarity, we selected three representative data-borrowing methods. And it will better highlight our method's performance trends.
>
>
>  > **Q5:** In Table 3, why is there no comparison of mean squared error (MSE) across different estimators? In several cases, it appears that the proposed estimator performs no better — or even worse — than the regular AIPW estimator.
>
> **Response to Q5:**  Thanks for your comments.
> **Here are the reasons why we don't  include MSE across different estimators:**
> - Limited space for presentation.
> - As the MSE between different estimates is already presented in Figure 3, it is not necessary to repeat it in Table 3.
>
> **For the comment that "in several cases, it appears that the proposed estimator performs no better than the regular AIPW estimator." We would like to kindly remind the reviewer that there may have been a misunderstanding about Figure 3. Below, we provide a detailed explanation.**
> - Figure 3 illustrates the trend of the MSE of $\hat\tau_{S}$  across a wide range of $K$ (corresponding to different $S$), rather than showing only the optimal $K$. Thus, **we do not require $\hat{\tau}_{S}$ to perform better for all values of $K$.**
> - From Figure 3, we can see that a better estimator can be obtained when $K$ is appropriately selected. This indicates that our method has great potential to improve upon previous approaches.
>
> > **Q6:** In the experiments, there are 300 treatment units and 100 controls, so it is reasonable to expect that borrowing additional external controls may be beneficial. However, how would the method perform if the number of controls already exceeds the number of treatment units — for example, 100 treatment units versus 300 controls?
>
> **Response to Q6:** Thanks for your comments. **First, we would like to clarify that the ratio between treatment and control (i.e., $N_t:N_c$) is not the most critical factor. Below, we provide an intuitive explanation of this.**
> - We focus on settings where the overall RCT data sample is relatively small (the most common cases in applications); this means that both the treatment and control groups in the RCT have smaller sample sizes relative to external control data.
> - In this setting, the ratio between treatment and control (i.e., $N_t:N_c$) is not the most critical factor. Logically speaking, the most critical factor is the number of comparable samples available in the external controls, compared with $N_c$.
>
> **Second, we conducted additional experiments to verify this assertion.** We explored different ratios $N_t:N_c$ and different numbers of external data $N_\mathcal{O}$, where the data-generating mechanisms remain the same in the manuscript. The experimental results are shown in the table below, where $K$ is the optimal value corresponding to the minimum MSE. The experimental results show that the method we proposed is consistently optimal.
>
> |$N_{\mathcal{O}}=800$| | | |
> |--|--|--|--|
> |MSE(optimal-K=400)|$N_t:N_c=3:1$|$N_t:N_c=1:1$|$N_t:N_c=1:3$|
> |$\hat{\tau}_{\textup{aipw}}$|0.0115|0.0122|0.0161|
> |$\hat{\tau}_{\textup{full}}$|0.0173|0.0205|0.0190|
> |$\hat{\tau}_{\textup{lasso}}$|0.0126|0.0123|0.0200|
> |$\hat{\tau}_{\textup{if}}$|**0.0066**| **0.0089**|**0.0131** |
> | | | | |
> | $N_{\mathcal{O}}=2000$| | | |
> |MSE(optimal-K=1000)|$N_t:N_c=3:1$| $N_t:N_c=1:1$ | $N_t:N_c=1:3$|
> |$\hat{\tau}_{\textup{aipw}}$|0.0275|0.0122| 0.0162|
> |$\hat{\tau}_{\textup{full}}$|0.0206|0.0135| 0.0111|
> |$\hat{\tau}_{\textup{lasso}}$|0.0492|0.0147| 0.0235|
> |$\hat{\tau}_{\textup{if}}$| **0.0104**|**0.0082**|**0.0073**|
> | | | | |
> |$N_{\mathcal{O}}=4000$| | | |
> |MSE(optimal-K=2000)|$N_t:N_c=3:1$|$N_t:N_c=1:1$|$N_t:N_c=1:3$|
> |$\hat{\tau}_{\textup{aipw}}$ |0.0275|0.0190|0.0162|
> |$\hat{\tau}_{\textup{full}}$ |0.0368|0.0288|0.0238|
> |$\hat{\tau}_{\textup{lasso}}$|0.0192|0.0134|0.0101|
> |$\hat{\tau}_{\textup{if}}$|**0.0073**|**0.0050**|**0.0042**|

---

> > ### Comment · Reviewer_NK6K · 2025-08-05
> > **Thanks!**
> >
> > I want to thank the authors for the clarifications. However, I still feel that most of the arguments are based simply on heuristics, lacking theoretical justifications. Therefore, I will keep my score.

---

> ### Author Response · Authors · 2025-08-05
> **Response to Official Comment by Reviewer NK6K**
>
> We sincerely thank the reviewers for their timely feedback.
>
> > **Q1:** I still feel that most of the arguments are based simply on heuristics, lacking theoretical justifications.
>
> **Response:** Thank you for your comment. **We respectfully disagree with the claim that our paper lacks theoretical justifications. Our work is supported by strong theoretical guarantees.  As recognized by other reviewers.**
> * "It presents **theoretically grounded**, empirically validated, and practically relevant methodology" (Reviewer ctUy);
> *  "This paper presnets **rigorous asymptotic analysis of bias and variance**" and "The influence-based approach is relatively novel, **theoretically grounded** ...." (Reviewer mm3d);
> * "the authors .... **analyzed its bias, and proved asymptotic normality of the estimation error.**" (Reviewer uy6o).
>
> Thus, our work is theoretically grounded.
>
> **In addition, we would like to clarify that our work tackles a highly challenging task: developing an individual-level adaptive data-borowwing framework that satisfies several notable (salient) advantages, as summarized in the table below.**
>
> |                                 | Don't rely on exchangeability assumption | Achieving sample-level sample selection  | Model-free to outcome model in external controls|Robust to outliers in external controls |
> |---------------------------------|----------------------------------|---------------------|------------------| ----------------|
> | Li et al. (2023, Biometrics) | ×   | × | × | × |
> | Gao et al. (2024, Biometrika) | √   | √ | × | × |
> | Ours | √   | √ | √ | √ |
>
>
> **This table highlights the novelty and unique contributions of our work.** We believe that our method tackles a highly challenging problem and offers meaningful contributions. **We are a bit unclear about the comment that 'the arguments are based simply on heuristics,' We would appreciate it if you could kindly elaborate on the comment, as more specific feedback would be very helpful for us to improve the paper.**
>
> **We hope that the above clarification addresses your concerns and sincerely invite you to kindly re-evaluate our work again. Thanks.**

---

> > ### Author Response · Authors · 2025-08-07
> > **Response to Official Comment by Reviewer NK6K (Additional Theoretical Analysis for MSE)**
> >
> > Dear Reviewer NK6K,
> >
> > I hope this message finds you well. In your last feedback, you mentioned that 'most of the arguments are based simply on heuristics.' **We greatly value your comments and, in response, have added several additional theoretical analyses to further support the proposed method.** Specifically, we first present the variance reduction achieved by leveraging the additional subset of external controls $\mathcal{S}$, as shown in Proposition R1 below.
> >
> > **Proposition R1.** Under Assumption 1 only,
> >  \$\\text{asy.var}(\\hat \\tau\_{\\mathcal{S}}) \\leq \\text{asy.var}(\\hat \\tau\_{\\textup{aipw}})\$. The variance reduction ( \$(N\_{\\mathcal{E}} + N\_{\\mathcal{S}} ) \\{ \\text{asy.var}(\\hat \\tau\_{\\textup{aipw}}) - \\text{asy.var}(\\hat \\tau\_{\\mathcal{S}})\\}\$) is\
> >  \\begin{align\*}   \\mathbb E\_{S} \\left \[ \\frac{ \\pi(X) }{q\^2} \\frac{\\text{Var}\_{S}(Y (0)\|X)}{ 1 - e_1(X) } \\frac{ \\mathbb P\_{S}(A=0, R=0\|X) }{ \\mathbb P\_{S}(A=0\|X) } \\right \],\ \\end{align\*}\
> >  where \$\\pi(X)\$, \$\\mathbb P\_{S}(A=0, R=0\|X) = \\mathbb P\_{S}(R=0\|X) \\mathbb P\_{S}(A=0\|R=0,X) = 1-\\pi(X)\$, \$\\mathbb P\_{S}(A=0\|X) = 1- e\_{S}(X)\$.
> >
> >
> > According to the result of Theorem 1, this variance reduction presented in Proposition R1 may come at the cost of increased bias given by $\mathbb{E}_{S}[ \frac{R}{q}( \mu_0(X) - m_0(X))]$. Thus, we have the following conclusion.
> >
> >
> > **Proposition R2.** Under Assumption 1, the asymptotic MSE of $\hat \tau_{S}$ is smaller than that of $\hat \tau_{aipw}$, provided that
> > \\\[ \\mathbb{E}\_{S} \\left \[ \\frac{ \\pi(X) }{q\^2} \\frac{\\text{Var}\_{S}(Y (0)\|X)}{ 1 - e_1(X) } \\frac{ \\mathbb{P}\_{S}(A=0, R=0\|X) }{ \\mathbb{P}\_{S}(A=0\|X) } \\right \] \> \| \\sqrt{N\_{\\mathcal{E}} + N\_{\\mathcal{S}}} \\cdot \\mathbb{E}\_{S}(\\frac{R}{q}( \\mu_0(X) - m_0(X))) \|\^2. \\\]
> >
> >
> > Proposition 2 provides the condition under which the estimator $\hat \tau_{S}$ has a smaller MSE than $\hat \tau_{\text{aipw}}$. It shows that only when the bias is sufficiently small can the additional use of external controls improve upon the estimator $\hat \tau_{\text{aipw}}$.
> >
> > **We hope that the above clarifications address your concerns.** If you have any further questions or concerns, we would be glad to discuss them with you. In addition, **we sincerely invite you, if possible, to kindly re-evaluate and support our work.** Thanks again.
> >
> > Warm regards,
> >
> > Authors

---

> > > ### Author Response · Authors · 2025-08-09
> > > **Thanks to Reviewer NK6K and Request for Further Feedback**
> > >
> > > Dear Reviewer NK6K,
> > >
> > > I hope this message finds you well.
> > >
> > > I’m writing to kindly follow up and ask whether your concerns have been fully addressed in our latest response. Your feedback has been invaluable to us, and we’ve made every effort to carefully incorporate your suggestions.
> > >
> > > Thank you again for your time and thoughtful input.
> > >
> > > Warm regards,
> > >
> > > Authors

---

### Decision · Program_Chairs · 2025-09-17

**Decision:**

Accept (poster)

**Comment:**

This paper proposes an influence-function-based framework for adaptively borrowing external control samples to improve treatment effect estimation in RCTs. The method assigns influence scores to external controls, develops a semiparametric efficient estimator, and uses a bias-variance tradeoff to select optimal subsets. Reviewers found the work theoretically grounded and practically relevant, noting its novelty in moving beyond global exchangeability assumptions and demonstrating robustness across scenarios. I encourage the authors to include a candid discussion of the concerns raised in the review process, including computational scalability, limited empirical scope, and the lack of formal treatment of post-selection inference variability. This will strengthen the paper and clarify the practical boundaries of the proposed method.